# Provable Tensor Factorization with Missing Data

**Prateek Jain**
Microsoft Research
Bangalore, India
prajain@microsoft.com

**Sewoong Oh**
Dept. of Industrial and Enterprise Systems Engineering
University of Illinois at Urbana-Champaign
Urbana, IL 61801
swoh@illinois.edu

## Abstract

We study the problem of low-rank tensor factorization in the presence of missing data. We ask the following question: how many sampled entries do we need, to efficiently and exactly reconstruct a tensor with a low-rank orthogonal decomposition? We propose a novel alternating minimization based method which iteratively refines estimates of the singular vectors. We show that under certain standard assumptions, our method can recover a three-mode $n \times n \times n$ dimensional rank-$r$ tensor exactly from $O(n^{3/2}r^5 \log^4 n)$ randomly sampled entries. In the process of proving this result, we solve two challenging sub-problems for tensors with missing data. First, in analyzing the initialization step, we prove a generalization of a celebrated result by Szemerédie et al. on the spectrum of random graphs. We show that this initialization step alone is sufficient to achieve the root mean squared error on the parameters bounded by $C(r^2 n^{3/2}(\log n)^4/|\Omega|)$ from $|\Omega|$ observed entries for some constant $C$ independent of $n$ and $r$. Next, we prove global convergence of alternating minimization with this good initialization. Simulations suggest that the dependence of the sample size on the dimensionality $n$ is indeed tight.

## 1 Introduction

Several real-world applications routinely encounter multi-way data with structure which can be modeled as low-rank tensors. Moreover, in several settings, many of the entries of the tensor are missing, which motivated us to study the problem of low-rank tensor factorization with missing entries. For example, when recording electrical activities of the brain, the electroencephalography (EEG) signal can be represented as a three-way array (temporal, spectral, and spatial axis). Oftentimes signals are lost due to mechanical failure or loose connection. Given numerous motivating applications, several methods have been proposed for this tensor completion problem. However, with the exception of 2-way tensors (i.e., matrices), the existing methods for higher-order tensors do not have theoretical guarantees and typically suffer from the curse of local minima.

In general, finding a factorization of a tensor is an NP-hard problem, even when all the entries are available. However, it was recently discovered that by restricting attention to a sub-class of tensors such as low-CP rank orthogonal tensors [1] or low-CP rank incoherent[1] tensors [2], one can efficiently find a provably approximate factorization. In particular, exact recovery of the factorization is possible for a tensor with a low-rank orthogonal CP decomposition [1]. We ask the question of recovering such a CP-decomposition when only a small number of entries are revealed, and show that exact reconstruction is possible even when we do not observe any entry in most of the fibers.

**Problem formulation.** We study tensors that have an orthonormal CANDECOMP/PARAFAC (CP) tensor decomposition with a small number of components. Moreover, for simplicity of notation and

exposition, we only consider symmetric third order tensors. We would like to stress that our techniques generalizes easily to handle *non-symmetric tensors* as well as *higher-order* tensors. Formally, we assume that the true tensor $T$ has the the following form:

$$T = \sum_{\ell=1}^{r} \sigma_\ell (\mathbf{u}_\ell \otimes \mathbf{u}_\ell \otimes \mathbf{u}_\ell) \in \mathbb{R}^{n \times n \times n} , \tag{1}$$

with $r \ll n$, $u_\ell \in \mathbb{R}^n$ with $\|u_\ell\| = 1$, and $u_\ell$'s are orthogonal to each other. We let $U \in \mathbb{R}^{n \times r}$ be a tall-orthogonal matrix where $u_\ell$'s is the $\ell$-th column of $U$ and $U_i \perp U_j$ for $i \neq j$. We use $\otimes$ to denote the standard outer product such that the $(i, j, k)$-th element of $T$ is given by: $T_{ijk} = \sum_a \sigma_a U_{ia} U_{ja} U_{ka}$. We further assume that the $u_i$'s are unstructured, which is formalized by the notion of *incoherence* commonly assumed in matrix completion problems. The *incoherence* of a symmetric tensor with orthogonal decomposition is

$$\mu(T) \equiv \max_{i \in [n], \ell \in [r]} \sqrt{n} |U_{i\ell}| , \tag{2}$$

where $[n] = \{1, \ldots, n\}$ is the set of the first $n$ integers. Tensor completion becomes increasingly difficult for tensors with larger $\mu(T)$, because the 'mass' of the tensor can be concentrated on a few entries that might not be revealed. Out of $n^3$ entries of $T$, a subset $\Omega \subseteq [n] \times [n] \times [n]$ is revealed. We use $\mathcal{P}_\Omega(\cdot)$ to denote the projection of a matrix onto the revealed set such that

$$\mathcal{P}_\Omega(T)_{ijk} = \begin{cases} T_{ijk} & \text{if } (i, j, k) \in \Omega , \\ 0 & \text{otherwise} . \end{cases}$$

We want to recover $T$ exactly using the given entries $(P_\Omega(T))$. We assume that each $(i, j, k)$ for all $i \leq j \leq k$ is included in $\Omega$ with a *fixed probability* $p$ (since $T$ is symmetric, we include all permutations of $(i, j, k)$). This is equivalent to fixing the total number of samples $|\Omega|$ and selecting $\Omega$ *uniformly at random* over all $\binom{n^3}{|\Omega|}$ choices. The goal is to ensure exact recovery with high probability and for $|\Omega|$ that is sub-linear in the number of entries $(n^3)$.

**Notations.** For a tensor $T \in \mathbb{R}^{n \times n \times n}$, we define a linear mapping using $U \in \mathbb{R}^{n \times m}$ as $T[U, U, U] \in \mathbb{R}^{m \times m \times m}$ such that $T[U, U, U]_{ijk} = \sum_{a,b,c} T_{abc} U_{ai} U_{bj} U_{ck}$. The spectral norm of a tensor is $\|T\|_2 = \max_{\|x\|=1} T[x, x, x]$. The Hilbert-Schmidt norm (Frobenius norm for matrices) of a tensor is $\|T\|_F = (\sum_{i,j,k} T_{ijk}^2)^{1/2}$. The Euclidean norm of a vector is $\|\mathbf{u}\|_2 = (\sum_i \mathbf{u}_i^2)^{1/2}$. We use $C, C'$ to denote any positive numerical constants and the actual value might change from line to line.

## 1.1 Algorithm

Ideally, one would like to minimize the rank of a tensor that explains all the sampled entries.

$$\underset{\widehat{T}}{\text{minimize}} \quad \text{rank}(\widehat{T}) \tag{3}$$

$$\text{subject to} \quad T_{ijk} = \widehat{T}_{ijk} \text{ for all } (i, j, k) \in \Omega .$$

However, even computing the rank of a tensor is NP-hard in general, where the rank is defined as the minimum $r$ for which CP-decomposition exists [3]. Instead, we fix the rank of $\widehat{T}$ by explicitly modeling $\widehat{T}$ as $\widehat{T} = \sum_{\ell \in [r]} \sigma_\ell (\mathbf{u}_\ell \otimes \mathbf{u}_\ell \otimes \mathbf{u}_\ell)$, and solve the following problem:

$$\underset{\widehat{T}, \text{rank}(\widehat{T})=r}{\text{minimize}} \left\| \mathcal{P}_\Omega(T) - \mathcal{P}_\Omega(\widehat{T}) \right\|_F^2 = \underset{\{\sigma_\ell, \mathbf{u}_\ell\}_{\ell \in [r]}}{\text{minimize}} \left\| \mathcal{P}_\Omega(T) - \mathcal{P}_\Omega\Big( \sum_{\ell \in [r]} \sigma_\ell (\mathbf{u}_\ell \otimes \mathbf{u}_\ell \otimes \mathbf{u}_\ell) \Big) \right\|_F^2 \tag{4}$$

Recently, [4, 5] showed that an alternating minimization technique can recover a matrix with missing entries exactly. We generalize and modify the algorithm for the case of higher order tensors and study it rigorously for tensor completion. However, due to special structure in higher-order tensors, our algorithm as well as analysis is significantly different than the matrix case (see Section 2.2 for more details).

To perform the minimization, we repeat the outer-loop getting refined estimates for all $r$ components. In the inner-loop, we loop over each component and solve for $\mathbf{u}_q$ while fixing the others $\{\mathbf{u}_\ell\}_{\ell \neq q}$.

More precisely, we set $\widehat{T} = \mathbf{u}_q^{t+1} \otimes \mathbf{u}_q \otimes \mathbf{u}_q + \sum_{\ell \neq q} \sigma_\ell \mathbf{u}_\ell \otimes \mathbf{u}_\ell \otimes \mathbf{u}_\ell$ in (4) and then find optimal $\mathbf{u}_q^{t+1}$ by minimizing the least squares objective given by (4). That is, each inner iteration is a simple least squares problem over the known entries, hence can be implemented efficiently and is also embarrassingly parallel.

---

**Algorithm 1** Alternating Minimization for Tensor Completion

---

1: Input: $P_\Omega(T), \Omega, r, \tau, \mu$
2: Initialize with $[(\mathbf{u}_1^0, \sigma_1), (\mathbf{u}_2^0, , \sigma_2), \ldots, (\mathbf{u}_r^0, \sigma_r)] = RTPM(P_\Omega(T), r)$         (RTPM of [1])
3: $[\mathbf{u}_1, \mathbf{u}_2, \ldots, \mathbf{u}_r] = \text{Threshold}([\mathbf{u}_1^0, \mathbf{u}_2^0, \ldots, \mathbf{u}_r^0], \mu)$         (Clipping scheme of [4])
4: **for all** $t = 1, 2, \ldots, \tau$ **do**
5:     /*OUTER LOOP */
6:     **for all** $q = 1, 2, \ldots, r$ **do**
7:         /*INNER LOOP*/
8:         $\hat{\mathbf{u}}_1^{t+1} = \arg\min_{\mathbf{u}_q^{t+1}} \|\mathcal{P}_\Omega(T - \mathbf{u}_q^{t+1} \otimes \mathbf{u}_q \otimes \mathbf{u}_q - \sum_{\ell \neq q} \sigma_\ell \cdot \mathbf{u}_\ell \otimes \mathbf{u}_\ell \otimes \mathbf{u}_\ell)\|_F^2$
9:         $\sigma_q^{t+1} = \|\hat{\mathbf{u}}_q^{t+1}\|_2$
10:         $\mathbf{u}_q^{t+1} = \hat{\mathbf{u}}_1^{t+1}/\|\hat{\mathbf{u}}_q^{t+1}\|_2$
11:     **end for**
12:     $[\mathbf{u}_1, \mathbf{u}_2, \ldots, \mathbf{u}_r] \leftarrow [\mathbf{u}_1^{t+1}, \mathbf{u}_2^{t+1}, \ldots, \mathbf{u}_r^{t+1}]$
13:     $[\sigma_1, \sigma_2, \ldots, \sigma_r] \leftarrow [\sigma_1^{t+1}, \sigma_2^{t+1}, \ldots, \sigma_r^{t+1}]$
14: **end for**
15: Output: $\widehat{T} = \sum_{q \in [r]} \sigma_q(\mathbf{u}_q \otimes \mathbf{u}_q \otimes \mathbf{u}_q)$

---

The *main novelty* in our approach is that we refine all $r$ components iteratively as opposed to the sequential deflation technique used by the existing methods for tensor decomposition (for fully observed tensors). In sequential deflation methods, components $\{\mathbf{u}_1, \mathbf{u}_2, \ldots, \mathbf{u}_r\}$ are estimated sequentially and estimate of say $\mathbf{u}_2$ is not used to refine $\mathbf{u}_1$. In contrast, our algorithm iterates over all $r$ estimates in the inner loop, so as to obtain refined estimates for all $\mathbf{u}_i$'s in the outer loop. We believe that such a technique could be applied to improve the error bounds of (fully observed) tensor decomposition methods as well.

As our method is directly solving a non-convex problem, it can easily get stuck in local minima. The key reason our approach can overcome the curse of local minima is that we start with a provably good initial point which is only a small distance away from the optima. To obtain such an initial estimate, we compute a low-rank approximation of the observed tensor using *Robust Tensor Power Method* (RTPM) [1]. RTPM is a generalization of the widely used power method for computing leading singular vectors of a matrix and can approximate the largest singular vectors up to the spectral norm of the "error" tensor. Hence, the challenge is to show that the error tensor has small spectral norm (see Theorem 2.1). We perform a thresholding step similar to [4] (see Lemma A.4) after the RTPM step to ensure that the estimates we get are incoherent.

Our analysis requires the sampled entries $\Omega$ to be independent of the current iterates $\mathbf{u}_i, \forall i$, which in general is not possible as $\mathbf{u}_i$'s are computed using $\Omega$. To avoid this issue, we divide the given samples ($\Omega$) into equal $r \cdot \tau$ parts randomly where $\tau$ is the number of outer loops (see Algorithm 1).

## 1.2 Main Result

**Theorem 1.1.** *Consider any rank-$r$ symmetric tensor $T \in \mathbb{R}^{n \times n \times n}$ with an orthogonal CP decomposition in (1) satisfying $\mu$-incoherence as defined in (2). For any positive $\varepsilon > 0$, there exists a positive numerical constant $C$ such that if entries are revealed with probability*

$$p \geq C \frac{\mu^6 r^5 \sigma_{\max}^4 (\log n)^4 \log(r\|T\|_F/\varepsilon)}{\sigma_{\min}^4 n^{3/2}},$$

*where $\sigma_{\max} \triangleq \max_\ell \sigma_\ell$ and $\sigma_{\min} \triangleq \min_\ell \sigma_\ell$, then the following holds with probability at least $1 - n^{-5} \log_2(4\sqrt{r}\|T\|_F/\varepsilon)$:*

- *the problem (3) has a unique optimal solution; and*

- $\log_2(\frac{4\sqrt{r}\|T\|_F}{\varepsilon})$ *iterations of Algorithm 1 produces an estimate $\widehat{T}$ s.t. $\|T - \widehat{T}\|_F \leq \varepsilon$ .*

The above result can be generalized to $k$-mode tensors in a straightforward manner, where exact recovery is guaranteed if, $p \geq C \frac{\mu^6 \, r^5 \, \sigma_{\max}^{2k-2} \, (\log n)^4 \, \log(r\|T\|_F/\varepsilon)}{\sigma_{\min}^4 \, n^{k/2}}$. However, for simplicity of notations and to emphasize key points of our proof, we only focus on 3-mode tensors in Section 2.3.

We provide a proof of Theorem 1.1 in Section 2. For an incoherent, well-conditioned, and low-rank tensor with $\mu = O(1)$ and $\sigma_{\min} = \Theta(\sigma_{\max})$, alternating minimization requires $O(r^5 n^{3/2}(\log n)^4)$ samples to get within an arbitrarily small normalized error. This is a vanishing fraction of the total number of entries $n^3$. Each step in the alternating minimization requires $O(r|\Omega|)$ operations, hence the alternating minimization only requires $O(r|\Omega| \log(r\|T\|_F/\varepsilon))$ operations. The initialization step requires $O(r^c|\Omega|)$ operations for some positive numerical constant $c$ as proved in [1]. When $r \ll n$, the computational complexity scales linearly in the sample size up to a logarithmic factor.

A fiber in a third order tensor is an $n$-dimensional vector defined by fixing two of the axes and indexing over remaining one axis. The above theorem implies that among $n^2$ fibers of the form $\{T[\mathbb{I}, e_j, e_k]\}_{j,k \in [n]}$, exact recovery is possible even if only $O(n^{3/2}(\log n)^4)$ fibers have non-zero samples, that is most of the fibers are not sampled at all. This should be compared to the matrix completion setting where all fibers are required to have at least one sample.

However, unlike matrices, the fundamental limit of higher order tensor completion is not known. Building on the percolation of Erdös-Renýi graphs and the coupon-collectors problem, it is known that matrix completion has multiple rank-$r$ solutions when the sample size is less than $C\mu r n \log n$ [6], hence exact recovery is impossible. But, such arguments do not generalize directly to higher order; see Section 2.5 for more discussion. Interestingly, simulations in Section 1.3 suggests that for $r = O(\sqrt{n})$, the sample complexity scales as $(r^{1/2}n^{3/2} \log n)$. That is, assuming the sample complexity provided by simulations is correct, our result achieves optimal dependence on $n$ (up to log factors). However, the dependency on $r$ is sub-optimal (see Section 2.5 for a discussion).

## 1.3 Empirical Results

Theorem 1.1 guarantees exact recovery when $p \geq Cr^5(\log n)^4/n^{3/2}$. Numerical experiments show that the average recovery rate converges to a universal curve over $\alpha$, where $p^* = \alpha r^{1/2} \ln n/((1-\rho)n^{3/2})$ in Figure 1. Our bound is tight in its dependency $n$ up to a poly-logarithmic factor, but is loose in its dependency in the rank $r$. Further, it is able to recover the original matrix exactly even when the factors are not strictly orthogonal.

We generate orthogonal matrices $U = [\mathbf{u}_1, \ldots, \mathbf{u}_r] \in \mathbb{R}^{n \times r}$ uniformly at random with $n = 50$ and $r = 3$ unless specified otherwise. For a rank-$r$ tensor $T = \sum_{i=1}^r \mathbf{u}_i \otimes \mathbf{u}_i \otimes \mathbf{u}_i$, we randomly reveal each entry with probability $p$. A tensor is exactly recovered if the normalized root mean squared error, $\text{RMSE} = \|T - \hat{T}\|_F/\|T\|_F$, is less than $10^{-72}$. Varying $n$ and $r$, we plot the recovery rate averaged over 100 instances as a function of $\alpha$. The degrees of freedom in representing a symmetric tensor is $\Omega(rn)$. Hence for large, $r$ we need number of samples scaling as $r$. Hence, the current dependence of $p^* = O(\sqrt{r})$ can only hold for $r = O(n)$. For not strictly orthogonal factors, the algorithm is robust. A more robust approach for finding an initial guess could improve the performance significantly, especially for non-orthogonal tensors.

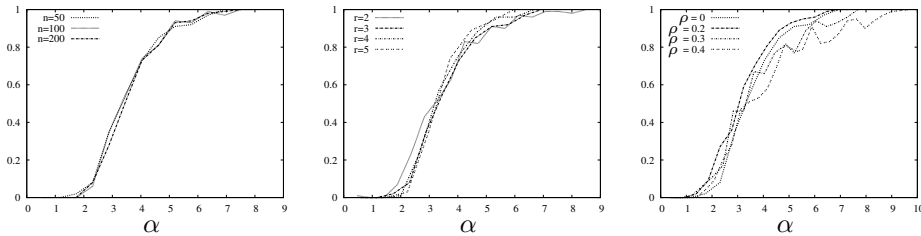

Figure 1: Average recovery rate converges to a universal curve over $\alpha$ when $p = \alpha r^{1/2} \ln n/((1-\rho)n^{3/2})$, where $\rho = \max_{i \neq j \in [r]} \langle \mathbf{u}_i, \mathbf{u}_j \rangle$ and $r = O(\sqrt{n})$.

## 1.4 Related Work

*Tensor decomposition and completion*: The CP model proposed in [7, 8, 9] is a multidimensional generalization of singular value decomposition of matrices. Computing the CP decomposition involves two steps: first apply a whitening operator to the tensor to get a lower dimensional tensor with orthogonal CP decomposition. Such a whitening operator only exists when $r \leq n$. Then, apply known power-method techniques for exact orthogonal CP decomposition [1]. We use this algorithm as well as the analysis for the initial step of our algorithm. For motivation and examples of orthogonal CP models we refer to [10, 1].

Recently, many heuristics for tensor completion have been developed such as the weighted least squares [11], Gauss-Newton [12], alternating least-squares [13, 14], trace norm minimization [15]. However, no theoretical guarantees are known for these approaches. In a different context, [16] shows that minimizing a weighted trace norm of flattened tensor provides exact recovery using $O(rn^{3/2})$ samples, but each observation needs to be a dense random projection of the tensor as opposed to observing just a single entry, which is the case in the tensor completion problem. In [17], an adaptive sampling method with an estimation algorithm was proposed that provably recovers a $k$-mode rank-$r$ tensor with $O(nr^{k-0.5}\mu^{k-1}k\log(r))$. However, the estimation algorithm as wells the analysis crucially relies on adaptive sampling and does not generalize to random samples.

*Relation to matrix completion*: Matrix completion has been studied extensively in the last decade since the seminal paper [18]. Since then, provable approaches have been developed, such as, nuclear norm minimization [18, 19], OptSpace [20, 21], and Alternating Minimization [4]. However, several aspects of tensor factorization makes it challenging to adopt matrix completion approaches directly. First, there is no natural convex surrogate of the tensor rank function and developing such a function is in fact a topic of active research [22, 16]. Next, even when all entries are revealed, tensor decomposition methods such as simultaneous power iteration are known to get stuck at local extrema, making it challenging to apply matrix decomposition methods directly. Third, for the initialization step, the best low-rank approximation of a matrix is unique and finding it is trivial. However, for tensors, finding the best low-rank approximation is notoriously difficult.

On the other hand, some aspects of tensor decomposition makes it possible to prove stronger results. Matrix completion aims to recover the underlying matrix only, since the factors are not uniquely defined due to invariance under rotations. However, for orthogonal CP models, we can hope to recover the individual singular vectors $\mathbf{u}_i$'s exactly. In fact, Theorem 1.1 shows that our method indeed recovers the individual singular vectors exactly.

*Spectral analysis of tensors and hypergraphs*: Theorem 2.1 and Lemma 2.2 should be compared to copious line of work on spectral analysis of matrices [23, 20], with an important motivation of developing fast algorithms for low-rank matrix approximations. We prove an analogous guarantee for higher order tensors and provide a fast algorithm for low-rank tensor approximation. Theorem 2.1 is also a generalization of the celebrated result of Friedman-Kahn-Szemerédi [24] and Feige-Ofek [25] on the second eigenvalue of random graphs. We provide an upper bound the largest second eigenvalue of a random hypergraph, where each edge includes three nodes and each of the $\binom{n}{3}$ edges is selected with probability $p$.

## 2 Analysis of the Alternating Minimization Algorithm

In this section, we provide a proof of Theorem 1.1 and the proof sketches of the required main technical theorems. We refer to the Appendix for formal proofs of the technical theorems and lemmas. There are two key components: $a$) the analysis of the initialization step (Section 2.1); and $b$) the convergence of alternating minimization given a sufficiently accurate initialization (Section 2.2). We use these two analyses to prove Theorem 1.1 in Section 2.3.

### 2.1 Initialization Analysis

We first show that $(1/p)\mathcal{P}_\Omega(T)$ is close to $T$ in spectral norm, and use it bound the error of robust power method applied directly to $\mathcal{P}_\Omega(T)$. The normalization by $(1/p)$ compensates for the fact that many entries are missing. For a proof of this theorem, we refer to Appendix A.

**Theorem 2.1** (Initialization). *For $p = \alpha/n^{3/2}$ satisfying $\alpha \geq \log n$, there exists a positive constant $C > 0$ such that, with probability at least $1 - n^{-5}$,*

$$\frac{1}{T_{\max} n^{3/2} p} \|\mathcal{P}_\Omega(T) - pT\|_2 \leq \frac{C (\log n)^2}{\sqrt{\alpha}} , \tag{5}$$

*where $T_{\max} \equiv \max_{i,j,k} T_{ijk}$, and $\|T\|_2 \equiv \max_{\|u\|=1} T[u,u,u]$ is the spectral norm.*

Notice that $T_{\max}$ is the maximum entry in the tensor $T$ and the factor $1/(T_{\max}n^{3/2}p)$ corresponds to normalization with the worst case spectral norm of $pT$, since $\|pT\|_2 \leq T_{\max}n^{3/2}p$ and the maximum is achieved by $T = T_{\max}(\mathbb{1} \otimes \mathbb{1} \otimes \mathbb{1})$. The following theorem guarantees that $O(n^{3/2}(\log n)^2)$ samples are sufficient to ensure that we get arbitrarily small error. A formal proof is provided in the Appendix.

Together with an analysis of *robust tensor power method* [1, Theorem 5.1], the next error bound follows from directly substituting (5) and using the fact that for incoherent tensors $T_{\max} \leq \sigma_{\max}\mu(T)^3 r/n^{3/2}$. Notice that the estimates can be computed efficiently, requiring only $O(\log r + \log\log \alpha)$ iterations, each iteration requiring $O(\alpha n^{3/2})$ operations. This is close to the time required to read the $|\Omega| \simeq \alpha n^{3/2}$ samples. One caveat is that we need to run robust power method $\text{poly}(r \log n)$ times, each with fresh random initializations.

**Lemma 2.2.** *For a $\mu$-incoherent tensor with orthogonal decomposition $T = \sum_{\ell=1}^{r} \sigma_\ell^*(\mathbf{u}_\ell^* \otimes \mathbf{u}_\ell^* \otimes \mathbf{u}_\ell^*) \in \mathbb{R}^{n \times n \times n}$, there exists positive numerical constants $C, C'$ such that when $\alpha \geq C(\sigma_{\max}/\sigma_{\min})^2 r^5 \mu^6 (\log n)^4$, running $C'(\log r + \log\log \alpha)$ iterations of the robust tensor power method applied to $\mathcal{P}_\Omega(T)$ achieves*

$$\|\mathbf{u}_\ell^* - \mathbf{u}_\ell^0\|_2 \leq C' \frac{\sigma_{\max}^*}{|\sigma_\ell^*|} \frac{\mu^3 r (\log n)^2}{\sqrt{\alpha}} ,$$

$$\frac{|\sigma_\ell^* - \sigma_\ell|}{|\sigma_\ell^*|} \leq C' \frac{\sigma_{\max}^*}{|\sigma_\ell^*|} \frac{\mu^3 r (\log n)^2}{\sqrt{\alpha}} ,$$

*for all $\ell \in [r]$ with probability at least $1 - n^{-5}$, where $\sigma_{\max}^* = \max_{\ell \in [r]} |\sigma_\ell^*|$ and $\sigma_{\min}^* = \min_{\ell \in [r]} |\sigma_\ell^*|$.*

## 2.2 Alternating Minimization Analysis

We now provide convergence analysis for the alternating minimization part of Algorithm 1 to recover rank-$r$ tensor $T$. Our analysis assumes that $\|\mathbf{u}_i - \mathbf{u}_i^*\|_2 \leq c\sigma_{\min}/r\sigma_{\max}, \forall i$ where $c$ is a small constant (dependent on $r$ and the condition number of $T$). The above mentioned assumption can be satisfied using our initialization analysis and by assuming $\Omega$ is large-enough.

At a high-level, our analysis shows that each step of Algorithm 1 ensures geometric decay of a distance function (specified below) which is "similar" to $\max_j \|\mathbf{u}_j^t - \mathbf{u}_j^*\|_2$.

Formally, let $T = \sum_{\ell=1}^{r} \sigma_\ell^* \cdot \mathbf{u}_\ell^* \otimes \mathbf{u}_\ell^* \otimes \mathbf{u}_\ell^*$. WLOG, we can assume that that $\sigma_\ell^* \leq 1$. Also, let $[U, \Sigma] = \{(\mathbf{u}_\ell, \sigma_\ell), 1 \leq \ell \leq r\}$, be the $t$-th step iterates of Algorithm 1. We assume that $\mathbf{u}_\ell^*, \forall \ell$ are $\mu$-incoherent and $\mathbf{u}_\ell, \forall \ell$ are $2\mu$-incoherent. Define, $\Delta_\ell^\sigma = \frac{|\sigma_\ell - \sigma_\ell^*|}{\sigma_\ell^*}$, $\mathbf{u}_\ell = \mathbf{u}_\ell^* + \mathbf{d}_\ell$, $(\Delta_\ell^\sigma)^{t+1} = \frac{|\sigma_\ell^{t+1} - \sigma_\ell^*|}{\sigma_\ell^*}$, and $\mathbf{u}_\ell^{t+1} = \mathbf{u}_\ell^* + \mathbf{d}_\ell^{t+1}$. Now, define the following distance function:

$$d_\infty([U, \Sigma], [U^*, \Sigma^*]) \equiv \max_\ell \left( \|\mathbf{d}_\ell\|_2 + \Delta_\ell^\sigma \right) .$$

The next theorem shows that this distance function decreases geometrically with number of iterations of Algorithm 1. A proof of this theorem is provided in Appendix B.4.

**Theorem 2.3.** *If $d_\infty([U, \Sigma], [U^*, \Sigma^*]) \leq \frac{1}{1600r} \frac{\sigma_{min}^*}{\sigma_{max}^*}$ and $\mathbf{u}_i$ is $2\mu$-incoherent for all $1 \leq i \leq r$, then there exists a positive constant $C$ such that for $p \geq \frac{Cr^2(\sigma_{\max}^*)^2 \mu^3 \log^2 n}{(\sigma_{\min}^*)^2 n^{3/2}}$ we have w.p. $\geq 1 - \frac{1}{n^7}$,*

$$d_\infty([U^{t+1}, \Sigma^{t+1}], [U^*, \Sigma^*]) \leq \frac{1}{2} d_\infty([U, \Sigma], [U^*, \Sigma^*]),$$

*where $[U^{t+1}, \Sigma^{t+1}] = \{(\mathbf{u}_\ell^{t+1}, \sigma_\ell^{t+1}), 1 \leq \ell \leq r\}$ are the $(t+1)$-th step iterates of Algorithm 1. Moreover, each $\mathbf{u}_\ell^{t+1}$ is $2\mu$-incoherent for all $\ell$.*

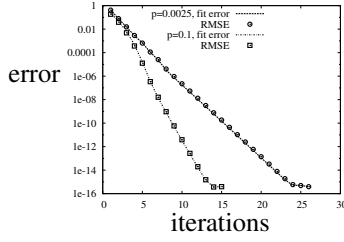

Figure 2: Algorithm 1 exhibits linear convergence until machine precision. For the estimate $\widehat{T}_t$ at the $t$-th iterations, the fit error $\|\mathcal{P}_\Omega(T - \widehat{T}_t)\|_F / \|\mathcal{P}_\Omega(T)\|_F$ closely tracks the normalized root mean squared error $\|T - \widehat{T}_t\|_F / \|T\|_F$, suggesting that it serves as a good stopping criterion.

Note that our number of samples depend on the number of iterations $\tau$. But due to linear convergence, our sample complexity increases only by a factor of $\log(1/\epsilon)$ where $\epsilon$ is the desired accuracy.

*Difference from Matrix AltMin*: Here, we would like to highlight differences between our analysis and analysis of the alternating minimization method for matrix completion (matrix AltMin) [4, 5]. In the matrix case, the singular vectors $\mathbf{u}_i^*$'s need not be unique. Hence, the analysis is required to guarantee a decay in the subspace distance $dist(U, U^*)$; typically, principal angle based subspace distance is used for analysis. In contrast, orthonormal $\mathbf{u}_i^*$'s uniquely define the tensor and hence one can obtain distance bounds $\|\mathbf{u}_i - \mathbf{u}_i^*\|_2$ for each component $\mathbf{u}_i$ individually.

On the other other hand, an iteration of the matrix AltMin iterates over all the vectors $\mathbf{u}_i, 1 \leq i \leq r$, where $r$ is the rank of the current iterate and hence don't have to consider the error in estimation of the fixed components $U_{[r]\backslash q} = \{\mathbf{u}_\ell, \forall \ell \neq q\}$, which is a challenge for the analysis of Algorithm 1 and requires careful decomposition and bounds of the error terms.

## 2.3 Proof of Theorem 1.1

Let $T = \sum_{q=1}^r \sigma_q^*(u_q^* \otimes u_q^* \otimes u_q^*)$. Denote the initial estimates $U^0 = [u_1^0, \ldots, u_r^0]$ and $\sigma^0 = [\sigma_1^0, \ldots, \sigma_r^0]$ to be the output of robust tensor power method at step 5 of Algorithm 1. With a choice of $p \geq C(\sigma_{\max}^*)^4 \mu^6 r^4 (\log n)^4 / (\sigma_{\min}^*)^4 n^{3/2}$ as per our assumption, Lemma 2.2 ensures that we have $\|u_q^0 - u_q^*\| \leq \sigma_{\min}^* / (4800\, r\sigma_{\max})$ and $|\sigma_q^0 - \sigma_q^*| \leq |\sigma_q^*|\sigma_{\min}^*/(4800\, r\sigma_{\max})$ with probability at least $1 - n^{-5}$. This requires running robust tensor power method for $(r \log n)^c$ random initializations for some positive constant $c$, each requiring $O(|\Omega|)$ operations ignoring logarithmic factors.

To ensure that we have sufficiently incoherent initial iterate, we perform thresholding proposed in [4]. In particular, we threshold all the elements of $\mathbf{u}_i^0$ (obtained from RTPM method, see Step 3 of Algorithm 1) that are larger (in magnitude) than $\mu/\sqrt{n}$ to be $\frac{sign(\mathbf{u}_\ell(i))\mu}{\sqrt{n}}$ and then re-normalize to obtain $\mathbf{u}_i$. Using Lemma A.4, this procedure ensures that the obtained initial estimate $\mathbf{u}_i$ satisfies the two criteria that is required by Theorem 2.3: a) $\|\mathbf{u}_i - \mathbf{u}_i^*\|_2 \leq \frac{1}{1600r} \cdot \frac{\sigma_{min}^*}{\sigma_{max}^*}$, and b) $\mathbf{u}_i$ is $2\mu$-incoherent.

With this initialization, Theorem 2.3 tells us that $O(\log_2(4r^{1/2}\|T\|_F/\varepsilon))$ iterations (each iteration requires $O(r|\Omega|)$ operations) is sufficient to achieve:

$$\|u_q - u_q^*\|_2 \leq \frac{\varepsilon}{4r^{1/2}\|T\|_F} \quad \text{and} \quad |\sigma_q - \sigma_q^*| \leq \frac{|\sigma_q^*|\varepsilon}{4r^{1/2}\|T\|_F} \ ,$$

for all $q \in [r]$ with probability at least $1 - n^{-7} \log_2(4r^{1/2}\|T\|_F/\varepsilon)$. The desired bound follows from the next lemma with a choice of $\tilde{\varepsilon} = \varepsilon/4r^{1/2}\|T\|_F$. For a proof we refer to Appendix B.6.

**Lemma 2.4.** *For an orthogonal rank-r tensor $T = \sum_{q=1}^r \sigma_q^*(u_q^* \otimes u_q^* \otimes u_q^*)$ and any rank-r tensor $\widehat{T} = \sum_{q=1}^r \sigma_q(u_q \otimes u_q \otimes u_q)$ satisfying $\|u - u^*\|_2 \leq \tilde{\varepsilon}$ and $|\sigma - \sigma^*| \leq |\sigma^*|\tilde{\varepsilon}$ for all $q \in [r]$ and for all positive $\tilde{\varepsilon} > 0$, we have $\|T - \widehat{T}\|_F \leq 4\, r^{1/2} \|T\|_F \tilde{\varepsilon}$.*

### 2.4 Fundamental limit and random hypergraphs

For matrices, it is known that exact matrix completion is impossible if the underlying graph is disconnected. For Erdös-Renýi graphs, when sample size is less than $C\mu rn \log n$, no algorithm can recover the original matrix [6]. However, for tensor completion and random hyper graphs, such a simple connection does not exist. It is not known how the properties of the hyper graph is related to recovery. In this spirit, a rank-one third-order tensor completion has been studied in a specific context of *MAX-3LIN problems*. Consider a series of linear equations over $n$ binary variables $x = [x_1 \ldots x_n] \in \{\pm 1\}^n$. An instance of a 3LIN problem consists of a set of linear equations on GF(2), where each equation involve exactly three variables, e.g.

$$x_1 \oplus x_2 \oplus x_3 = +1 \,, \quad x_2 \oplus x_3 \oplus x_4 = -1 \,, \quad x_3 \oplus x_4 \oplus x_5 = +1 \tag{6}$$

We use $-1$ to denote true (or 1 in GF(2)) and $+1$ to denote false (or 0 in GF(2)). Then the exclusive-or operation denoted by $\oplus$ is the integer multiplication. the MAX-3LIN problem is to find a solution $x$ that satisfies as many number of equations as possible. This is an NP-hard problem in general, and hence random instances of the problem with a *planted solution* has been studied [26]. Algorithm 1 provides a provable guarantee for MAX-3LIN with random assignments.

**Corollary 2.5.** *For random MAX-3LIN problem with a planted solution, under the hypotheses of Theorem 1.1, Algorithm 1 finds the correct solution with high probability.*

Notice that this tensor has incoherence one and rank one. This implies exact reconstruction for $P \geq C(\log n)^4/n^{3/2}$. This significantly improves over a message-passing approach to MAX-3LIN in [26], which is guaranteed to find the planted solution for $p \geq C(\log \log n)^2/(n \log n)$. It was suggested that a new notion of connectivity called *propagation connectivity* is a sufficient condition for the solution of random MAX-3LIN problem with a planted solution to be unique [26, Proposition 2]. Precisely, it is claimed that if the hypergraph corresponding to an instance of MAX-3LIN is propagation connected, then the optimal solution for MAX-3LIN is unique and there is an efficient algorithm that finds it. However, the example in 6 is propagation connected but there is no unique solution: both $[1, 1, 1, -1, -1]$ and $[1, -1, -1, 1, -1]$ satisfy the equations. Hence, propagation connectivity is not a sufficient condition for uniqueness of the MAX-3LIN solution.

### 2.5 Open Problems and Future Directions

*Tensor completion for non-orthogonal decomposition.* Numerical simulations suggests that non-orthogonal CP models can be recovered exactly (without the usual whitening step). It would be interesting to analyze our algorithm under non-orthogonal CP model. However, we would like to point here that even with fully observed tensor, exact factorization is known only for orthonormal tensors. Now, given that our method guarantees not only completion but also tensor factorization (which is essential for large scale applications), our method would require a similar condition.

*Optimal dependence on $r$.* The numerical results suggest the threshold sample size scaling as $\sqrt{r}$. This is surprising since the degrees of freedom in describing a CP model scales linearly in $r$, implying that the $\sqrt{r}$ scaling only holds for $r = O(\sqrt{n})$. In comparison, for matrix completion the threshold scales as $r$. It is important to understand why this change in dependence in $r$ happens for higher order tensors, and identify how it depends on $k$ for $k$-th order tensor completion.

*Mis-specified $r$ and $\mu$.* The algorithm requires the knowledge of the rank $r$ and the incoherence $\mu$. The algorithm is not sensitive to the knowledge of $\mu$. In fact, all the numerical experiments are run without specifying the incoherence, and without the clipping step. An interesting direction is to understand the price of mis-specified rank and to estimate the true rank from data.

## Footnotes

[1]The notion of incoherence we assume in (2) can be thought of as incoherence between the fibers and the standard basis vectors.

[2] A MATLAB implementation of Algorithm 1 used to run the experiments is available at http://web.engr.illinois.edu/~swoh/software/optspace.

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
