[Supplementary Material]

# Supplementary Material for Provable Tensor Factorization with Missing Data

## A   Proof of Theorem 2.1 for Initialization Analysis

We prove the following bound on the spectrum of random tensors:

$$\max_{x,y,z,\|x\|=\|y\|=\|z\|=1} \big(\mathcal{P}_\Omega(T) - p\,T\big)[x,y,z] \quad \leq \quad C\,T_{\max}\,(\log n)^2\,\sqrt{(n_1\,n_2\,n_3)^{1/2}\,p}\;.$$

Here we prove the theorem for general case where $T$ is not symmetric and might even have different dimensions $n_1$, $n_2$ and $n_3$. Inspired by [24, 20], our strategy is as follows:

(1) Reduce to $x$,$y$, and $z$ which belongs to *discretized sets* $\widetilde{S}_{n_1}$, $\widetilde{S}_{n_2}$, and $\widetilde{S}_{n_3}$;

(2) Bound the contribution of *light triples* using concentration of measure;

(3) Bound the contribution of *heavy triples* using the discrepancy property of a random tripartite hypergraph.

Define a discretization of an $n$-dimensional ball as

$$\widetilde{S}_n \quad \equiv \quad \Big\{ x \in \Big\{\frac{\Delta}{\sqrt{n}}\mathbb{Z}\Big\}^n \; : \; \|x\| \leq 1 \Big\},$$

such that $\widetilde{S}_n \subseteq S_n \equiv \{x \in \mathbb{R}^n \; : \; \|x\| \leq 1\}$. Later we will set $\Delta$ to be a small enough constant.

**Lemma A.1** (Remark 4.1 in [20]). *For any tensor $A \in \mathbb{R}^{n_1 \times n_2 \times n_3}$,*

$$\max_{x \in S_{n_1}, y \in S_{n_2}, x \in S_{n_3}} A[x,y,z] \quad \leq \quad \max_{x \in \widetilde{S}_{n_1}, y \in \widetilde{S}_{n_2}, x \in \widetilde{S}_{n_3}} \frac{1}{(1-\Delta)^2} A[x,y,z]$$

It is therefore enough show that the bound holds for discretized vectors all discretized vectors $x$, $y$, and $z$. One caveat is that such a probabilistic bound must hold with probability sufficiently close to one such that we can apply the union bound over all discretized choices of $x$, $y$, and $z$. The following lemma bounds the number of such choices.

**Lemma A.2** ([20]). *The size of the discretized set is bounded by $|\widetilde{S}_n| \leq (\Delta/10)^n$.*

A naive approach to upper bound $(\mathcal{P}_\Omega(T) - p\,T)[x,y,z]$ would be to consider it as a random variable and apply concentration inequalities directly. However, this naive approach fails since $x$, $y$ and $z$ can contain entries that are much larger than their typical value of $O(1/\sqrt{n})$. We thus separate the analysis into two contributions, and apply concentration inequalities to bound the contribution of the *light triples* and use graph topology of the random sampling to bound the contribution of the *heavy triples*. Define the light triples as

$$\mathcal{L} \quad \equiv \quad \Big\{ (i,j,k) \; : \; \big|T_{ijk}x_i y_j z_k\big| \leq T_{\max}\sqrt{\frac{\epsilon}{n_1\,n_2\,n_3}} \Big\}\;. \tag{7}$$

Heavy triples are defined as its complement $\overline{\mathcal{L}} = \{[n_1] \times [n_2] \times [n_3]\} \setminus \mathcal{L}$. Later we will set the appropriate value for $\epsilon = \Theta(p\sqrt{n_1 n_2 n_3})$. We can then write each contributions separately as

$$\Big|\big(\mathcal{P}_\Omega(T) - p\,T\big)[x,y,z]\Big| \quad \leq \quad \Big|\sum_{(i,j,k)\in\mathcal{L}} \big(\mathcal{P}_\Omega(T)_{ijk}x_i y_j z_k\big) - p\,T[x,y,z]\Big| + \Big|\sum_{(i,j,k)\in\overline{\mathcal{L}}} \mathcal{P}_\Omega(T)_{ijk}x_i y_j z_k\Big| \tag{8}$$

We will prove that both contributions are upper bounded by $CT_{\max}(\log n)^2\sqrt{(n_1 n_2 n_3)^{1/2}p}$ with some positive constant $C$ for all $x \in \widetilde{S}_{n_1}$, $y \in \widetilde{S}_{n_2}$, and $z \in \widetilde{S}_{n_3}$. The bound on the light triples follows from Chernoff's concentration inequalities. The bound on the heavy triples follows from the discrepancy property of random hyper graphs, which implies that there cannot be too many triples with large contributions. Theorem 2.1 then follows from Lemma A.1 with an appropriate choice of $\Delta = \Theta(1)$.

## A.1 Bounding the contribution of light triples

Let $Z \equiv \sum_{(i,j,k)\in\mathcal{L}} \left( \mathcal{P}_\Omega(T)_{ijk} x_i y_j z_k \right) - p\, T[x,y,z]$ for some $x \in S_{n_1}$, $y \in S_{n_2}$, and $z \in S_{n_3}$. We claim that

$$\mathbb{P}\left( Z > \frac{p T_{\max} \sqrt{n_1 n_2 n_3}}{\sqrt{\epsilon}} + t\left(n_1 + n_2 + n_3\right) \frac{2 T_{\max}\sqrt{\epsilon}}{\sqrt{n_1 n_2 n_3}} \right) \leq \exp\left\{ -t(n_2 + n_2 + n_3) \right\} . \quad (9)$$

We first show that the mean of $Z$ is bounded as

$$\left| \mathbb{E}[Z] \right| \leq 2\, p\, T_{\max} \sqrt{\frac{n_1 n_2 n_3}{\epsilon}} . \quad (10)$$

The mean can be written as
$\mathbb{E}[Z] = p \sum_{\mathcal{L}} T_{ijk} x_i y_j z_k - p \sum_{[n_1]\times[n_2]\times[n_3]} T_{ijk} x_i y_j z_k = p \sum_{\overline{\mathcal{L}}} T_{ijk} x_i y_j z_k$. Using the fact that for heavy triples $|T_{ink} x_i y_j z_k| \geq T_{\max} \sqrt{\epsilon/(n_1 n_2 n_3)}$, the expected contribution is then bounded by

$$\left| \sum_{(i,j,k)\in\overline{\mathcal{L}}} T_{ijk} x_i y_j z_k \right| \leq \sum_{(i,j,k)\in\overline{\mathcal{L}}} \frac{T_{ijk}^2 x_i^2 y_j^2 z_k^2}{|T_{ijk} x_i y_j z_k|}$$

$$\leq \frac{\sqrt{n_1 n_2 n_3}}{T_{\max}\sqrt{\epsilon}} \sum_{(i,j,k)\in\overline{\mathcal{L}}} T_{ijk}^2 x_i^2 y_j^2 z_k^2$$

$$\leq \frac{T_{\max}\sqrt{n_1 n_2 n_3}}{\sqrt{\epsilon}} .$$

We next show concentration of $Z$ around item mean. Let $\lambda = \sqrt{n_1 n_2 n_3}/(2 T_{\max}\sqrt{\epsilon})$ such that $|\lambda T_{ijk} x_i y_j z_k| \leq 1/2$ for all $(i,j,k) \in \mathcal{L}$. Then,
$e^{\lambda T_{ijk} x_i y_j z_k} - 1 \leq \lambda T_{ijk} x_i y_j z_k + 2\lambda(T_{ijk} x_i y_j z_k)^2$.

$$\mathbb{E}[e^{\lambda Z}] = \exp\{-\lambda p\, T[x,y,z]\} \prod_{(i,j,k)\in\mathcal{L}} \left( 1 - p + p\, e^{\lambda T_{ijk} x_i y_j z_k} \right)$$

$$\leq \exp\{-\lambda p\, T[x,y,z]\} \prod_{(i,j,k)\in\mathcal{L}} \left( 1 + p\left(\lambda T_{ijk} x_i y_j z_k + 2\lambda^2 (T_{ijk} x_i y_j z_k)^2\right) \right)$$

$$\leq \exp\left\{ p \sum_{\mathcal{L}} \lambda T_{ijk} x_i y_j z_k - \lambda p\, T[x,y,z] + p \sum_{\mathcal{L}} 2\lambda^2 (T_{ijk} x_i y_j z_k)^2 \right\}$$

$$\leq \exp\left\{ \lambda \mathbb{E}[Z] + \frac{p\, n_1 n_2 n_3}{2\epsilon} \right\} .$$

Applying Chernoff bound $\mathbb{P}(Z - \mathbb{E}[Z] > t) \leq \mathbb{E}[e^{\lambda Z}] e^{-\lambda \mathbb{E}[Z] - \lambda t}$, this proves (9). Note that the deviation of $-Z$ can be bounded similarly. We can now finish the proof of upper bound on the contribution of light triples by taking the union bound over all discretized vectors inside the ball. Setting $t = 2\log(20/\Delta)$ in (9), we get

$$\mathbb{P}\left( \max_{x\in\widetilde{S}_{n_1}, y\in\widetilde{S}_{n_2}, z\in\widetilde{S}_{n_3}} \sum_{(i,j,k)\in\mathcal{L}} T_{ijk} x_i y_j z_k - p\, T[x,y,z] \geq \frac{p T_{\max}\sqrt{n_1 n_2 n_3}}{\sqrt{\epsilon}} + 2\log(20/\Delta)\left(n_1 + n_2 + n_3\right) \frac{2 T_{\max}\sqrt{\epsilon}}{\sqrt{n_1 n_2 n_3}} \right)$$

$$\leq 2\, e^{(n_1+n_2+n_3)\log(20/\Delta)}\, e^{-2\log(20/\Delta)(n_1+n_2+n_3)}$$

$$\leq 2 e^{-(n_1+n_2+n_3)\log(20/\Delta)} .$$

Since $p = \epsilon/\sqrt{n_1 n_2 n_3}$, this proves that the contribution of light triples is bounded by $C\, T_{\max}\sqrt{\epsilon}$ with high probability.

Note that for the range of $p = \epsilon/n^2$, the contribution of light couples is bounded by $C\, T_{\max}\sqrt{\epsilon/n}$. However, even in this regime of $p$, the contribution of heavy triples is still $\Omega(1)$, which dominates the light triples by a factor of $\sqrt{n}$. This is the reason for the choice of $p = \Theta(\epsilon/n^{1.5})$.

## A.2 Bounding the contribution of heavy triples

The contribution of heavy triples is bounded by

$$\left| \sum_{(i,j,k) \in \overline{\mathcal{L}}} T_{ijk} x_i y_j z_k \right| \leq T_{\max} \sum_{(i,j,k) \in \overline{\mathcal{L}}} |x_i y_j z_k| \ .$$

In the following, we will show that the right-hand side of the above inequality is upper bounded by

$$\sum_{(i,j,k) \in \overline{\mathcal{L}}} |x_i y_j z_k| \leq C \sqrt{\epsilon} (\log n)^2 \ ,$$

for some positive numerical constant $C > 0$ with probability larger than $1 - n^{-5}$.

We consider a hypergraph $G = ([n_1] \times [n_2] \times [n_3], E)$ with undirected hyper edges, where each edge connects three nodes, each one from each set $[n_1]$, $[n_2]$, and $[n_3]$. Given a sampling of entries in a tensor, we let the edges in $G$ denote the positions of the entries that is sampled. The proof is a generalization of similar proof for matrices in [24, 25, 20] and is based on two properties of the hypergraph $G$. Define the degree of a node as the number of edges connected to that particular node such that $\deg_1(i) \equiv |\{(i,j,k) \in E\}|$, and similarly define $\deg_2(j)$ and $\deg_3(k)$. Define the degree of two nodes as the number of edges connected to both of the nodes such that $\deg_{12}(i,j) \equiv |\{(i,j,k) \in E\}|$, and similarly define $\deg_{13}(i,k)$ and $\deg_{23}(j,k)$.

1. *Bounded degree property.* A hyper graph $G$ satisfies the bounded degree property if the degree are upper bounded as follows:

$$
\begin{aligned}
\deg_1(i) &\leq \xi_0 \, p \, n_2 n_3 & \text{for all } i \in [n_1] \ , \\
\deg_2(j) &\leq \xi_0 \, p \, n_1 n_3 & \text{for all } j \in [n_2] \ , \\
\deg_3(k) &\leq \xi_0 \, p \, n_1 n_2 & \text{for all } k \in [n_3] \ , \\
\deg_{12}(i,j) &\leq \xi_0 \, (p \, n_3 + \log n_3) & \text{for all } i \in [n_1], j \in [n_2] \ , \\
\deg_{13}(i,k) &\leq \xi_0 \, (p \, n_2 + \log n_2) & \text{for all } i \in [n_1], k \in [n_3] \ , \\
\deg_{23}(j,k) &\leq \xi_0 \, (p \, n_1 + \log n_1) & \text{for all } j \in [n_2], k \in [n_3] \ , \quad (11)
\end{aligned}
$$

   for some positive numerical constant $\xi_0 > 0$ (independent of $n_1, n_2, n_3$ and $p$) where $p = |E|/(n_1 n_2 n_3)$.

2. *Discrepancy property.* A hyper graph $G$ satisfies the discrepancy property if for any subset of nodes $A_1 \in [n_1]$, $A_2 \in [n_2]$, and $A_3 \in [n_3]$, at least one of the following is true:

$$e(A_1, A_2, A_3) \leq \xi_1 \, \bar{e}(A_1, A_2, A_3) \ , \qquad (12)$$

$$e(A_1, A_2, A_3) \ln \left( \frac{e(A_1, A_2, A_3)}{\bar{e}(A_1, A_2, A_3)} \right) \leq \xi_2 \max \left\{ |A_1| \ln \left( \frac{e \, n_1}{|A_1|} \right) , \ |A_2| \ln \left( \frac{e \, n_2}{|A_2|} \right) , \ |A_3| \ln \left( \frac{e \, n_3}{|A_3|} \right) \right\} (13)$$

   for some positive numerical constants $\xi_1, \xi_2 > 0$ (independent of $n_1, n_2, n_3$ and $p$). Here, $e(A_1, A_2, A_3)$ denotes the number of edges between the three subsets $A_1$, $A_2$ and $A_3$, and $\bar{e}(A_1, A_2, A_3) \equiv p |A_1| |A_2| |A_3|$ denotes the average number of edges between the three subsets.

We first prove that if the sampling pattern is defined by a graph $G$ which satisfies both the bounded degree and discrepancy properties, then the contribution of heavy triples is $O(\sqrt{\epsilon})$. Notice that this is a deterministic statement, that holds for all graphs with the above properties. We then finish the proof by showing that the random sampling satisfies both the bounded degree and discrepancy properties with probability at least $1 - n^{-5}$.

We partition the indices according to the value of corresponding vectors:

$$
\begin{aligned}
A_1^{(u)} &\equiv \left\{ i \in [n_1] \ : \ \frac{\Delta}{\sqrt{n_1}} 2^{u-1} \leq |x_i| < \frac{\Delta}{\sqrt{n_1}} 2^u \right\} \\
A_2^{(v)} &\equiv \left\{ j \in [n_2] \ : \ \frac{\Delta}{\sqrt{n_2}} 2^{v-1} \leq |y_j| < \frac{\Delta}{\sqrt{n_2}} 2^v \right\} \\
A_3^{(w)} &\equiv \left\{ k \in [n_3] \ : \ \frac{\Delta}{\sqrt{n_3}} 2^{w-1} \leq |z_k| < \frac{\Delta}{\sqrt{n_3}} 2^w \right\} \ ,
\end{aligned}
$$

for $u \in \{1, \ldots, \lceil \log_2(\sqrt{n_1}/\Delta) \rceil + 1\}$, $v \in \{1, \ldots, \lceil \log_2(\sqrt{n_2}/\Delta) \rceil + 1\}$, and $w \in \{1, \ldots, \lceil \log_2(\sqrt{n_3}/\Delta) \rceil + 1\}$. We denote the size of each set by $a_i^{(u)} \equiv |A_i^{(u)}|$. We use $e_{uvw}$ to denote the number of edges between three subsets $A_1^{(u)}$, $A_2^{(v)}$, and $A_3^{(w)}$, and we use $\bar{e}_{uvw} \equiv p \, a_1^{(u)} a_2^{(v)} a_3^{(w)}$ to denote the average number of edges. Notice that the above definition of $A_1^{(u)}$'s cover all non-zero values of the entries of $x$, since, with discretization, the smallest possible positive value is $\Delta/\sqrt{n_1}$. The same applies to the entries of $y$ and $z$.

$$
\begin{aligned}
\sum_{(i,j,k) \in \overline{\mathcal{L}}} \left| x_i y_j z_k \right| &\leq \sum_{(i,j,k):|x_i y_j z_k| > \sqrt{\epsilon/(n_1 n_2 n_3)}} \left| x_i y_j z_k \right| \\
&\leq \sum_{(u,v,w):2^{u+v+w} > 8\sqrt{\epsilon}/\Delta^3} e_{uvw} \underbrace{\frac{\Delta 2^u}{\sqrt{n_1}} \frac{\Delta 2^v}{\sqrt{n_2}} \frac{\Delta 2^w}{\sqrt{n_3}}}_{\sigma_{uvw}} .
\end{aligned}
$$

Note that since $\sum_u a_1^{(u)} 2^{2(u-1)} \Delta^2/n_1 \leq \|x\|^2 \leq 1$, we get that

$$
\begin{aligned}
a_1^{(u)} &\leq (n_1/\Delta^2) 2^{-2(u-1)} , \\
a_2^{(v)} &\leq (n_2/\Delta^2) 2^{-2(v-1)} , \\
a_3^{(w)} &\leq (n_3/\Delta^2) 2^{-2(w-1)} .
\end{aligned}
\tag{14}
$$

The contributions from various combinations of $(u, v, w)$ utilize various subsets of our assumptions. We prove that in each case the contribution is $O(\sqrt{\epsilon}(\log n)^2)$ as follows.

**Case1.** For $(u, v, w)$ satisfying the first discrepancy property (12) : $e_{uvw} \leq \xi_1 \bar{e}_{uvw}$.

In this case, using (14) and the fact that $p = \epsilon/\sqrt{n_1 n_2 n_3}$,

$$
\begin{aligned}
\sum \sigma_{uvw} &\leq \xi_1 p a_1^{(u)} a_2^{(v)} a_3^{(w)} \frac{\Delta^3 2^{u+v+w}}{\sqrt{n_1 n_2 n_3}} \\
&\leq \frac{64 \xi_1 p \sqrt{n_1 n_2 n_3}}{\Delta^3 2^{u+v+w}} \\
&\leq 16 \xi_1 \sqrt{\epsilon}(\log n)^2 ,
\end{aligned}
$$

where $n \equiv \max\{n_1, n_2, n_3\}$ and in the last inequality we used the fact that we are summing over heavy triples satisfying $\Delta^3 2^{u+v+w} > 8\sqrt{\epsilon}$, and $\sum_{(u,v,w):2^{u+v+w} \leq 8\sqrt{\epsilon}/\Delta^3} 2^{-(u+v+w)} \leq 2 \log_2(\sqrt{n_1}/\Delta) \log_2(\sqrt{n_2}/\Delta) \Delta^3/(8\sqrt{\epsilon})$.

**Case2.** For $(u, v, w)$ satisfying the second discrepancy property in (13).

**Case 2-1.** For $(u, v, w)$ satisfying $\ln(e_{uvw}/\bar{e}_{uvw}) \leq (1/2) \ln(en_3/a_3^{(w)}) = (1/4)(\ln(en_3/(a_3^{(w)} 2^{2w})) + \ln(2^{2w}))$.

**Case 2-1-1.** When $\ln(2^{2w}) \leq \ln(en_3/(a_3^{(w)} 2^{2w}))$, we have $\ln(e_{uvw}/\bar{e}_{uvw}) \leq \ln(en_2/(a_3^{(w)} 2^{2w}))$, which gives

$$
\begin{aligned}
e_{uvw} &\leq en_3 \bar{e}_{uvw}/(a_3^{(w)} 2^{2w}) \\
&\leq e \, a_1^{(u)} a_2^{(v)} n_3 p 2^{-2w} \\
&\leq \frac{16e}{\Delta^4} n_1 n_2 n_3 p 2^{-2(u+v+w)} .
\end{aligned}
$$

It follows that $\sum \sigma_{uvw} \leq (16/\Delta) p \sqrt{n_1 n_2 n_3} 2^{-u-v-w} \leq 2\Delta^2 \sqrt{\epsilon}(\log n)^2$ using the fact that we are summing over heavy triples.

**Case 2-1-2.** When $\ln(2^{2w}) > \ln(en_3/(a_3^{(w)} 2^{2w}))$, we have $\ln(e_{uvw}/\bar{e}_{uvw}) \leq \ln(2^w)$.

**Case 2-1-2-1.** For $\sqrt{\epsilon}e_{uvw} > 2^{u+v+w}\bar{e}_{uvw}$, it follows that $2^{u+v} \leq \sqrt{\epsilon}$. Since we are in the case where the first discrepancy does not hold, i.e. $e_{uvw} > \xi_1\bar{e}_{uvw}$, and the the second discrepancy property holds, we have $e_{uvw} \leq e_{uvw}\ln(e_{uvw}/\bar{e}_{uvw}) \leq \xi_2 a_3^{(w)}\ln(en_3/a_3^{(w)}) \leq 2\xi_2 a_3^{(w)}\ln(2^{2w})$ Then,

$$
\begin{aligned}
\sum \sigma_{uvw} &\leq \sum 2\xi_2 a_3^{(w)}\ln(2^{2w})\frac{\Delta^3 2^{u+v+w}}{\sqrt{n_1 n_2 n_3}} \\
&\leq \sum \frac{8\xi_2 \Delta\sqrt{n_3}2^{u+v}}{\sqrt{n_1 n_2}}\frac{\ln(2^w)}{2^w} \\
&\leq 8\Delta\xi_2\sqrt{\frac{n_3}{n_1 n_2}}\sqrt{\epsilon}\log_2(\sqrt{n_1}/\Delta)\log_2(\sqrt{n_3}/\Delta) ,
\end{aligned}
$$

which is $O\left(\sqrt{\epsilon}(\log n)^2\sqrt{(n_3/(n_1 n_2))}\right)$

**Case 2-1-2-2.** For $\sqrt{\epsilon}e_{uvw} \leq 2^{u+v+w}\bar{e}_{uvw}$,

$$
\begin{aligned}
\sum \sigma_{uvw} &\leq \sum \frac{2^{u+v+w}p\,a_1^{(u)}a_2^{(v)}a_3^{(w)}}{\sqrt{\epsilon}}\frac{\Delta^3 2^{u+v+w}}{\sqrt{n_1 n_2 n_3}} \\
&\leq \frac{\sqrt{\epsilon}}{\Delta^3}\sum \frac{a_1^{(u)}a_2^{(v)}a_3^{(w)}\Delta^3 2^{2(u+v+w)}}{n_1 n_2 n_3} \\
&\leq \frac{\sqrt{\epsilon}}{\Delta^3}\|x\|^2\|y\|^2\|z\|^2 ,
\end{aligned}
$$

which is $O(\sqrt{\epsilon})$.

**Case 2-2.** For $(u,v,w)$ satisfying $\ln(e_{uvw}/\bar{e}_{uvw}) > (1/2)\ln(en_3/a_3^{(w)})$.

**Case 2-2-1.** For $2^{u+v} \leq \sqrt{n_1 n_2\epsilon/n_3}2^w$, we know from the condition $\ln(e_{uvw}/\bar{e}_{uvw}) > (1/2)\ln(en_3/a_3^{(w)})$, that $e_{uvw} \leq 2\xi_2 a_3^{(w)}$. Then,

$$
\begin{aligned}
\sum \sigma_{uvw} &\leq \sum 2\xi_2 a_3^{(w)}\frac{2^{u+v+w}\Delta^3}{\sqrt{n_1 n_2 n_3}} \\
&\leq \sum 8\xi_2\Delta 2^{u+v-w}\sqrt{\frac{n_3}{n_1 n_2}} \\
&\leq 8\xi_2\Delta\sqrt{\epsilon}\log_2(\sqrt{n_1}/\Delta)\log_2(\sqrt{n_2}/\Delta) ,
\end{aligned}
$$

which is $O(\sqrt{\epsilon}(\log n)^2)$.

**Case 2-2-2.** For $2^{u+v} > \sqrt{n_1 n_2\epsilon/n_3}2^w$

**Case 2-2-2-1.** For $(u,v,w)$ satisfying bounded degree property with $\deg_{12}(i,j) \leq \xi_0 pn_3$, we have $e_{uvw} \leq a_1 a_2\xi_0 pn_3$. Then,

$$
\begin{aligned}
\sum \sigma_{uvw} &\leq \sum \frac{16\xi_0\epsilon}{\Delta}2^{w-u-v} \\
&\leq \sqrt{\frac{n_3}{n_1 n_2}}\frac{16\xi_0\sqrt{\epsilon}}{\Delta}\log_2(\sqrt{n_2}/\Delta)\log_2(\sqrt{n_2}/\Delta) ,
\end{aligned}
$$

which is $O(\sqrt{\epsilon}\sqrt{n_3/(n_1 n_2)}(\log_2 n)^2)$.

**Case 2-2-2-2.** For $(u,v,w)$ satisfying bounded degree property with $\deg_{12}(i,j) \leq \xi_0\log n_3$, we have $e_{uvw} \leq a_1 a_2\xi_0\log n$.

$$
\begin{aligned}
\sum \sigma_{uvw} &\leq \sum \frac{16\xi_0\log n_3 2^{w-u-v}}{\Delta}\sqrt{\frac{n_1 n_2}{n_3}} \\
&\leq \frac{16\xi_0\log n_3}{\Delta\sqrt{\epsilon}}\log_2(\sqrt{n_1}/\Delta)\log_2(\sqrt{n_2}/\Delta) ,
\end{aligned}
$$

which is $O((1/\sqrt{\epsilon})(\log n)^3)$.

For $\epsilon \geq \log n$, this proves that the contribution of the heavy triples is $O(\sqrt{\epsilon}(\log n)^2)$.

We are left to prove that the bounded degree and the bounded discrepancy properties hold for a random tripartite hypergraph $G = (V_1 \cup V_2 \cup V_3, E)$ where each edge is selected with probability $p$. Precisely, let $n = \max\{|V_1|, |V_2|, |V_3|\}$, then the following lemma provides a bound on the degree and discrepancy properties, with high probability.

**Lemma A.3.** *For any $\delta \in [0, 1/e]$ and $p \geq (1/n^2) \log n$, there exists numerical constants $C, C' > 0$ such that a random tripartite hyper graph satisfies the bounded degree property: for all $i \in V_1$, $j \in V_2$, and $k \in V_3$,*

$$
\begin{aligned}
\deg_1(i) &\leq 2pn_2 n_3 + \frac{8}{3} \log \frac{3n_1}{\delta} \\
\deg_2(j) &\leq 2pn_1 n_3 + \frac{8}{3} \log \frac{3n_2}{\delta} \\
\deg_3(k) &\leq 2pn_1 n_2 + \frac{8}{3} \log \frac{3n_3}{\delta} \\
\deg_{12}(i, j) &\leq 2pn_3 + \frac{8}{3} \log \frac{3n_1 n_2}{\delta} \\
\deg_{13}(i, k) &\leq 2pn_2 + \frac{8}{3} \log \frac{3n_1 n_3}{\delta} \\
\deg_{23}(j, k) &\leq 2pn_1 + \frac{8}{3} \log \frac{3n_2 n_3}{\delta}
\end{aligned}
$$

*and the bounded discrepancy property: for all subsets $A_1 \subseteq V_1$, $A_2 \subseteq V_2$, and $A_3 \subseteq V_3$, at least one of the following is true.*

$$
e(A_1, A_2, A_3) \leq C\alpha^2 \bar{e}(A_1, A_2, A_3)\left(1 + \frac{\ln(1/\delta)}{pn^2}\right), \text{ or}
$$

$$
e(A_1, A_2, A_3) \ln\left(\frac{e(A_1, A_2, A_3)}{\bar{e}(A_1, A_2, A_3)}\right) \leq C'\left(\ln\frac{\alpha}{\delta} + \max\left\{|A_1|\ln\frac{e\,n_1}{|A_1|}, \ |A_2|\ln\frac{e\,n_2}{|A_2|}, \ |A_3|\ln\frac{e\,n_3}{|A_3|}\right\}\right),
$$

*where $n_1 = |V_1|$, $n_2 = |V_2|$, $n_3 = |V_3|$, $n = \max\{n_1, n_2, n_2\}$ and $\alpha \equiv \max n_i/n_j$.*

Now, for the choice of $\delta = n^{-5}$, the bounded degree and discrepancy properties in (11), (12), and (13) hold for random tripartite hypergraphs. This finishes the proof of Theorem 2.1.

## A.3 Proof of the bounded degree and discrepancy properties in Lemma A.3

We first prove the bounded degree properties of (11) hold with probability at least $1 - \delta$. Applying standard concentration inequality, e.g. Bernstein inequality, we get that for some positive constant $\delta > 0$,

$$
\begin{aligned}
\mathbb{P}\left(\deg_1(i) \leq 2pn_2 n_3 + \frac{8}{3} \log \frac{3n_1}{\delta}\right) &\leq \exp\left(-\frac{(1/2)(pn_2 n_3 + (8/3)\log(3n_1/\delta))^2}{(1/3)(pn_2 n_3 + (8/3)\log(3n_1/\delta)) + n_2 n_3 p(1-p)}\right) \\
&\leq e^{-\log(3n_1/\delta)},
\end{aligned}
$$

for $n$ sufficiently large, and taking union bound over all choices of $i$, $j$ and $k$, $\deg_1(i)$, $\deg_2(j)$, and $\deg_3(k)$'s are uniformly bounded with probability at least $1 - \delta/2$.

Similarly, we can apply concentration inequality to bound for some positive constant $\delta > 0$

$$
\begin{aligned}
\mathbb{P}\left(\deg_{12}(i, j) \leq 2pn_3 + \frac{8}{3} \log \frac{3n_1 n_2}{\delta}\right) &\leq \exp\left(-\frac{(1/2)(pn_3 + (8/3)\log(3n_1 n_2/\delta))^2}{(1/3)(pn_3 + (8/3)\log(3n_1 n_2/\delta)) + n_3 p(1-p)}\right) \\
&\leq e^{-\log(3n_1 n_2/\delta)}.
\end{aligned}
$$

Applying the union bound over all choices of $(i, j)$, $(i, k)$ and $(j, k)$, we get that the bound holds uniformly with probability at least $1 - \delta/2$.

Next, we prove that the random hyper graphs satisfy the discrepancy properties of (12) and (13). For any given subsets $A_1 \subseteq [n_1]$, $A_2 \subseteq [n_2]$, and $A_2 \subseteq [n_3]$, let $a_1, a_2$, and $a_3$ denote the cardinality of the subsets, and $\bar{e}(A_1, A_2, A_3) = pa_1 a_2 a_3$.

Let's assume, without loss of generality, that $a_1 \leq a_2 \leq a_3$. We divide the analysis into two cases depending on the size of the smallest subset. When at least two of the subsets are large, i.e. $a_2 = \Omega(n)$ and $a_3 = \Omega(n)$, then by bounded degree property, we can prove that (12) holds. However, when $a_1$ and $a_2$ are small, e.g. $O(1)$, then the first discrepancy no longer holds, and we need a different technique to show concentration.

**Case 1.** When $a_1 \geq n_1/e$.

From the bounded degree property, we know that $\deg_1(i) \leq 2pn_2n_3 + (8/3)\ln(3n_1/\delta)$. Then,

$$
\begin{aligned}
e(A_1, A_2, A_3) &\leq a_1(2pn_2n_3 + (8/3)\ln(3n_1/\delta)) \\
&\leq a_1(5pn_2n_3 + (8/3)\ln(1/\delta)) \\
&\leq 5a_1 pn_2n_3 \left(1 + \frac{\ln(1/\delta)}{pn_2n_3}\right) \\
&\leq 5\,e^2\,\alpha^2\bar{e}(A_1, A_2, A_3)\left(1 + \frac{\ln(1/\delta)}{pn_2n_3}\right).
\end{aligned}
$$

**Case 2.** When $a_1 < n_1/e$.

We use the following bound on sum of indicator variables deviating from the mean :

$$
\mathbb{P}\big(e(A_1, A_2, A_3) \geq t\bar{e}(A_1, A_2, A_3)\big) \leq e^{-(1/3)\,\bar{e}\,t\ln t}, \tag{15}
$$

where we denote $\bar{e}(A_1, A_2, A_3)$ by $\bar{e}$, which holds for $t \geq 4$. For the bound holds with probability at least $1 - \delta$, we require

$$
e^{-(1/3)\bar{e}t\ln t}\binom{n_1}{a_1}\binom{n_2}{a_2}\binom{n_3}{a_3} \leq \frac{\delta}{n_1n_2n_3},
$$

where the term $1/(n_1n_2n_3)$ is chosen to compensate for the union bound over all choices of $a_1$, $a_2$ and $a_3$. Simplifying the combinatorial terms, we get

$$
e^{-(1/3)\bar{e}t\ln t}\left(\frac{en_1}{a_1}\right)^{a_1}\left(\frac{en_2}{a_2}\right)^{a_2}\left(\frac{en_3}{a_3}\right)^{a_3}e^{\ln(n_1n_2n_3/\delta)} \leq 1.
$$

Equivalently,

$$
a_1\ln\left(\frac{en_1}{a_1}\right) + a_2\ln\left(\frac{en_2}{a_2}\right) + a_3\ln\left(\frac{en_3}{a_3}\right) + \ln\left(\frac{n_1n_2n_3}{\delta}\right) \leq \frac{\bar{e}t\ln t}{3}.
$$

We assumed that $a_1 \leq n_1/e$, and since $x\ln(n_1/x)$ is monotone in $x \in [1, n_1/e]$, we know that $a_1\ln(en_1/a_1) \geq \ln n_1$.

$$
4a_1\ln\left(\frac{en_1}{a_1}\right) + a_2\ln\left(\frac{en_2}{a_2}\right) + a_3\ln\left(\frac{en_3}{a_3}\right) + \ln\left(\frac{\alpha^2}{\delta}\right) \leq \frac{\bar{e}t\ln t}{3}.
$$

To lighten the notations, let's suppose $a_1\ln(e\,n_1/a_1) \leq a_2\ln(e\,n_2/a_2) \leq a_3\ln(e\,n_3/a_3)$. Let $t'$ be the smallest number such that $(3/\bar{e})\big(6a_3\ln(e\,n_3/a_3) + \ln(\alpha^2/\delta)\big) = t'\ln t'$.

For the regime of parameters such that $t' \leq 4$, then $e(A_1, A_2, A_3) \leq 4\bar{e}(A_1, A_2, A_3)$ with probability at least $1 - \delta$ the bounded discrepancy condition, in particular the first one, holds.

For the regime of parameters such that $t' > 4$, we can apply (15) to get that with probability at least $1 - \delta$, the following holds uniformly for all choices of $A_1$, $A_2$, and $A_3$:

$$
e(A_1, A_2, A_3) \leq t'\bar{e}(A_1, A_2, A_3).
$$

Since we defined $t'$ to satisfy $\bar{e}t'\ln t' = 18a_3\ln(e\,n_3/a_3) + 3\ln(\alpha^2/\delta)$, we have

$$
e(A_1, A_2, A_3)\ln t' \leq 18a_3\ln(e\,n_3/a_3) + 3\ln(\alpha^2/\delta).
$$

As $t'$ upper bounds $e(A_1, A_2, A_3)/\bar{e}$, we have

$$
e(A_1, A_2, A_3)\ln\left(\frac{e(A_1, A_2, A_3)}{\bar{e}(A_1, A_2, A_3)}\right) \leq 18a_3\ln(e\,n_3/a_3) + 3\ln(\alpha^2/\delta).
$$

## A.4   Proof of Thresholding

**Lemma A.4.** *Let* $\mathbf{u}_\ell, 1 \leq \ell \leq r$ *be such that* $\|\mathbf{u}_\ell - \mathbf{u}_\ell^*\|_2 \leq \alpha$ *where* $\alpha < 1/4$. *Also, let* $\mathbf{u}_\ell^*, 1 \leq \ell \leq r$ *be* $\mu$-*incoherent unit vectors. Now define* $\widetilde{\mathbf{u}_\ell}$ *as:*

$$\widetilde{\mathbf{u}_\ell}(i) = \begin{cases} \mathbf{u}_\ell(i) & \text{if } |\mathbf{u}_\ell(i)| \leq \frac{\mu}{\sqrt{n}}, \\ sign(\mathbf{u}_\ell(i)) \frac{\mu}{\sqrt{n}} & \text{if } |\mathbf{u}_\ell(i)| > \frac{\mu}{\sqrt{n}}. \end{cases}$$

*Also, let* $\widehat{\mathbf{u}_\ell} = \widetilde{\mathbf{u}_\ell}/\|\widetilde{\mathbf{u}_\ell}\|_2$. *Then,* $\|\widehat{\mathbf{u}_\ell} - \mathbf{u}_\ell^*\|_2 \leq 3\alpha \ \forall 1 \leq \ell \leq r$ *and each* $\widehat{\mathbf{u}_\ell}$ *is* $2\mu$-*incoherent.*

*Proof.* As $\|\mathbf{u}_\ell^*\|_\infty \leq \frac{\mu}{\sqrt{n}}$, hence $\|\widetilde{\mathbf{u}_\ell} - \mathbf{u}_\ell^*\|_2 \leq \|\mathbf{u}_\ell - \mathbf{u}_\ell^*\|_2 \leq \alpha$, $\forall \ell$. This also implies that $1 - \alpha \leq \|\widetilde{\mathbf{u}_\ell}\|_2 \leq 1$. Hence,

$$\|\widehat{\mathbf{u}_\ell} - \mathbf{u}_\ell^*\|_2 \leq \|\widetilde{\mathbf{u}_\ell} - \mathbf{u}_\ell^*\|_2 + \left( \frac{1}{\|\widetilde{\mathbf{u}_\ell}\|_2} - 1 \right) \leq 3\alpha.$$

Moreover, $\|\widehat{\mathbf{u}_\ell}\|_\infty \leq \frac{\mu}{\sqrt{n}\cdot(1-\alpha)} \leq \frac{2\mu}{\sqrt{n}}$. Hence proved. □

# B   Alternating Minimization Analysis

## B.1   Main theorem for rank-two analysis

In this section, we provide convergence analysis for Algorithm 1 for the special case of a rank-2 orthonormal tensor $T$ with equal singular values, i.e. $T = \mathbf{u}_1^* \otimes \mathbf{u}_1^* \otimes \mathbf{u}_1^* + \mathbf{u}_2^* \otimes \mathbf{u}_2^* \otimes \mathbf{u}_2^*$, where $\mathbf{u}_1^*, \mathbf{u}_2^* \in \mathbb{R}^n$ are orthonormal vectors satisfying $\mu$-incoherence, i.e., $\|\mathbf{u}_i^*\|_\infty \leq \mu/\sqrt{n}$. The purpose of this example is to highlight the proof ideas and we fix $\sigma_1, \sigma_2$ to be both one at each step of Algorithm 1 for simplicity. The following theorem proves the desired linear convergence. Let $[\mathbf{u}_1^t, \mathbf{u}_2^t]$ denote the current estimate at the $t$-th iteration of Algorithm 1. For brevity, we drop the superscript indexing time and let $[\mathbf{u}_1, \mathbf{u}_2]$ denote $[\mathbf{u}_1^t, \mathbf{u}_2^t]$ whenever it is clear from the context.

**Theorem B.1.** *If* $\mathbf{u}_1$ *and* $\mathbf{u}_2$ *are* $2\mu$-*incoherent, then there exists a positive constant* $C$ *such that for* $p \geq C\frac{\mu^3 \log^2 n}{n^{1.5}}$ *the following holds (w.p.* $\geq 1 - \log(1/\epsilon)/n^8$):

$$d_\infty \left( [\mathbf{u}_1^{t+1}, \mathbf{u}_2^{t+1}], [\mathbf{u}_1^*, \mathbf{u}_2^*] \right) \leq \frac{1}{4} d_\infty \left( [\mathbf{u}_1, \mathbf{u}_2], [\mathbf{u}_1^*, \mathbf{u}_2^*] \right),$$

*where* $d_\infty([\mathbf{u}_1, \mathbf{u}_2], [\mathbf{u}_1^*, \mathbf{u}_2^*]) = \max_{i, 1 \leq i \leq 2} \|\mathbf{u}_i - \mathbf{u}_i^*\|_2$. *Moreover,* $\mathbf{u}_1^{t+1}$, $\mathbf{u}_2^{t+1}$ *are both* $2\mu$-*incoherent.*

*Proof.* We claim that with probability at least $1 - 1/n^8$,

$$\|\mathbf{u}_i^{t+1} - \mathbf{u}_i^*\|_2 \leq \frac{1}{4} \cdot d_\infty \left( [\mathbf{u}_1, \mathbf{u}_2], [\mathbf{u}_1^*, \mathbf{u}_2^*] \right),$$

for both $i \in \{1, 2\}$. This proves the desired bound. Incoherence of $[\mathbf{u}_1^{t+1}, \mathbf{u}_2^{t+1}]$ follows from Lemma B.2. Without loss of generality, we only prove the claim for $i = 1$. Recall that $\widehat{\mathbf{u}}_1^{t+1}$ is the solution of the least squares problem in Step 11 of Algorithm 1, and can be written as

$$\widehat{\mathbf{u}}_1^{t+1}(i) = \frac{\sum_{jk} \delta_{ijk} \mathbf{u}_1(j) \mathbf{u}_1(k) \mathbf{u}_1^*(j) \mathbf{u}_1^*(k)}{\sum_{jk} \delta_{ijk} (\mathbf{u}_1(j))^2 (\mathbf{u}_1(k))^2} \mathbf{u}_1^*(i) + \frac{\sum_{jk} \delta_{ijk} \mathbf{u}_1(j) \mathbf{u}_1(k) (\mathbf{u}_2^*(i) \mathbf{u}_2^*(j) \mathbf{u}_2^*(k) - \mathbf{u}_2(i) \mathbf{u}_2(j) \mathbf{u}_2(k))}{\sum_{jk} \delta_{ijk} (\mathbf{u}_1(j))^2 (\mathbf{u}_1(k))^2}.$$
(16)

Note that the update that can be written in a vector form:

$$\widehat{\mathbf{u}}^{t+1} = \langle \mathbf{u}_1, \mathbf{u}_1^* \rangle^2 \mathbf{u}_1^* + \langle \mathbf{u}_1, \mathbf{u}_2^* \rangle^2 \mathbf{u}_2^* - \langle \mathbf{u}_1, \mathbf{u}_2 \rangle^2 \mathbf{u}_2 + B^{-1}(\langle \mathbf{u}_1, \mathbf{u}_1^* \rangle^2 B - C) \mathbf{u}_1^*$$
$$+ B^{-1}(\langle \mathbf{u}_1, \mathbf{u}_2^* \rangle^2 B - F) \mathbf{u}_2^* - B^{-1}(\langle \mathbf{u}_1, \mathbf{u}_2 \rangle^2 B - G) \mathbf{u}_2, \quad (17)$$

where $B, C, F, G$ are all diagonal matrices, s.t.,

$$B_{ii} = \sum_{jk} \delta_{ijk} (\mathbf{u}_1(j))^2 (\mathbf{u}_1(k))^2, \quad C_{ii} = \sum_{jk} \delta_{ijk} \mathbf{u}_1(j) \mathbf{u}_1(k) \mathbf{u}_1^*(j) \mathbf{u}_1^*(k),$$

$$F_{ii} = \sum_{jk} \delta_{ijk} \mathbf{u}_1(j) \mathbf{u}_1(k) \mathbf{u}_2^*(j) \mathbf{u}_2^*(k), \quad G_{ii} = \sum_{jk} \delta_{ijk} \mathbf{u}_1(j) \mathbf{u}_1(k) \mathbf{u}_2(j) \mathbf{u}_2(k) . \quad (18)$$

Let $\widehat{\mathbf{u}}_1^{t+1} - \langle \mathbf{u}_1, \mathbf{u}_1^* \rangle^2 \mathbf{u}_1^* = \text{err}^0 + \text{err}^1 + \text{err}^2$, such that

$$\text{err}^0 = \langle \mathbf{u}_1, \mathbf{u}_2^* \rangle^2 \mathbf{u}_2^* - \langle \mathbf{u}_1, \mathbf{u}_2 \rangle^2 \mathbf{u}_2,$$
$$\text{err}^1 = B^{-1}(\langle \mathbf{u}_1, \mathbf{u}_1^* \rangle^2 B - C)\mathbf{u}^*,$$
$$\text{err}^2 = B^{-1}(\langle \mathbf{u}_1, \mathbf{u}_2^* \rangle^2 B - F)\mathbf{u}_2^* - B^{-1}(\langle \mathbf{u}_1, \mathbf{u}_2 \rangle^2 B - G)\mathbf{u}_2 . \quad (19)$$

We separate the analysis for each of the error terms. Using Lemma B.3, we have:

$$\|\text{err}^0\|_2 \leq 4 d_\infty\left([\mathbf{u}_1, \mathbf{u}_2], [\mathbf{u}_1^*, \mathbf{u}_2^*]\right) \|\mathbf{u}_2 - \mathbf{u}_2^*\|_2. \quad (20)$$

Setting $p \geq C \frac{\mu^3 \log^2 n}{\gamma^2 n^{3/2}}$ for a $\gamma$ to be chosen appropriately later and using Lemma B.7 and Lemma B.5, we have (w.p. $\geq 1 - 2/n^9$):

$$\|\text{err}^1\|_2 \leq \frac{\gamma}{1 - \gamma} \|\mathbf{u}_1 - \mathbf{u}_1^*\|_2. \quad (21)$$

Similarly, using Lemma B.4 and $p \geq C \frac{\mu^3 \log^2 n}{\gamma^2 n^{3/2}}$, we have (w.p. $\geq 1 - 1/n^9$):

$$\|\text{err}^2\| \leq 8 \frac{\gamma}{1 - \gamma} \cdot \|\mathbf{u}_2 - \mathbf{u}_2^*\|_2. \quad (22)$$

We want to upper bound the error:

$$\|\widehat{\mathbf{u}}_1^{t+1} - \mathbf{u}_1^*\|_2 \leq \|\widehat{\mathbf{u}}_1^{t+1} - \langle \mathbf{u}_1, \mathbf{u}_1^* \rangle^2 \mathbf{u}_1^*\|_2 + \|(\langle \mathbf{u}_1, \mathbf{u}_1^* \rangle^2 - 1)\mathbf{u}_1^*\|_2 .$$

Since $1 - \langle \mathbf{u}_1, \mathbf{u}_1^* \rangle^2 = (1/2)\|\mathbf{u}_1 - \mathbf{u}_1^*\|_2^2$, we have from (20), (21), and (22) that (w.p. $\geq 1 - 10/n^9$):

$$\|\widehat{\mathbf{u}}_1^{t+1} - \mathbf{u}_1^*\|_2 \leq \left(\frac{\gamma}{1 - \gamma} + \frac{\|\mathbf{u}_1 - \mathbf{u}_1^*\|_2}{2}\right) \|\mathbf{u}_1 - \mathbf{u}_1^*\|_2 + \left(8\frac{\gamma}{1 - \gamma} + 4 d_\infty\left([\mathbf{u}_1, \mathbf{u}_2], [\mathbf{u}_1^*, \mathbf{u}_2^*]\right)\right) \|\mathbf{u}_2 - \mathbf{u}_2^*\|_2 .$$

Setting $\gamma \leq 1/200$ and for $d_\infty\left([\mathbf{u}_1, \mathbf{u}_2], [\mathbf{u}_1^*, \mathbf{u}_2^*]\right) \leq 1/200$ as per our assumption, this proves the desired bound. $\qquad \square$

## B.2 Technical lemmas for rank-two analysis

The next lemma shows that all our estimates are $2\mu$-incoherent, which in turn allows us to bound the error in the above proof effectively. Note that the incoherence of the updates do not increase beyond a global constant ($2\mu$). Let $\widehat{\mathbf{u}}_1^{t+1}$ be obtained by update (16) and let $\mathbf{u}_1^{t+1} = \widehat{\mathbf{u}}_1^{t+1}/\|\widehat{\mathbf{u}}_1^{t+1}\|_2$.

**Lemma B.2.** *Under the hypotheses of Theorem B.1, $\mathbf{u}_1^{t+1}$ is $2\mu$-incoherent with probability at least $1 - 1/n^9$.*

*Proof.* Using (16) and the definitions of $B, C, F, G$ given in (18), we have:

$$|\widehat{\mathbf{u}}_1^{t+1}(i)| \leq \frac{|C_{ii}|}{|B_{ii}|}\frac{\mu}{\sqrt{n}} + \frac{|F_{ii}|}{|B_{ii}|}\frac{\mu}{\sqrt{n}} + \frac{|G_{ii}|}{|B_{ii}|}\frac{2\mu}{\sqrt{n}} \leq \frac{2\mu}{\sqrt{n}} , \quad (23)$$

where the second inequality follows by bounds on $B_{ii}, C_{ii}, F_{ii}, G_{ii}$ obtained using Lemma B.5 and the distance bound $d_\infty\left([\mathbf{u}_1, \mathbf{u}_2], [\mathbf{u}_1^*, \mathbf{u}_2^*]\right)$. $\qquad \square$

Next, we bound the first error term in (17).

**Lemma B.3.** *Let $\mathbf{u} = \mathbf{u}^* + \mathbf{d}^u$ and $\mathbf{v} = \mathbf{v}^* + \mathbf{d}^v$, where $\mathbf{u}, \mathbf{u}^*, \mathbf{v}, \mathbf{v}^*$ are all unit vectors and $\mathbf{u}^* \perp \mathbf{v}^*$. Also, let $\|\mathbf{d}^u\|_2 \leq 1$ and $\|\mathbf{d}^v\|_2 \leq 1$. Then, the following holds:*

$$\|\langle \mathbf{u}, \mathbf{v}^* \rangle^2 \mathbf{v}^* - \langle \mathbf{u}, \mathbf{v} \rangle^2 \mathbf{v}\| \leq 4(\|\mathbf{d}^u\|_2 + \|\mathbf{d}^v\|_2)\|\mathbf{d}^v\|_2.$$

*Proof.* Note that,

$$\langle \mathbf{u}, \mathbf{v} \rangle^2 = (\langle \mathbf{u}, \mathbf{v}^* \rangle + \langle \mathbf{u}, \mathbf{d}^v \rangle)^2 = \langle \mathbf{u}, \mathbf{v}^* \rangle^2 + \langle \mathbf{u}, \mathbf{d}^v \rangle^2 + 2\langle \mathbf{u}, \mathbf{v}^* \rangle \langle \mathbf{u}, \mathbf{d}^v \rangle. \quad (24)$$

Hence,

$$\|\langle \mathbf{u}, \mathbf{v}^* \rangle^2 \mathbf{v}^* - \langle \mathbf{u}, \mathbf{v} \rangle^2 \mathbf{v}\|_2 = \|\langle \mathbf{u}, \mathbf{v} \rangle^2 \mathbf{d}^v - \langle \mathbf{u}, \mathbf{d}^v \rangle^2 \mathbf{v}^* - 2\langle \mathbf{u}, \mathbf{v}^* \rangle \langle \mathbf{u}, \mathbf{d}^v \rangle \mathbf{v}^*\|_2,$$
$$\leq \langle \mathbf{u}, \mathbf{v} \rangle^2 \|\mathbf{d}^v\|_2 + \|\mathbf{d}^v\|^2 + 2|\langle \mathbf{u}, \mathbf{v}^* \rangle| \|\mathbf{d}^v\|_2. \quad (25)$$

Now, $\langle \mathbf{u}, \mathbf{v}^* \rangle = \langle \mathbf{d}^u, \mathbf{v}^* \rangle \leq \|\mathbf{d}^u\|_2$. Also,
$\langle \mathbf{u}, \mathbf{v} \rangle^2 \leq \langle \mathbf{u}, \mathbf{v} \rangle = (\langle \mathbf{d}^u, \mathbf{v}^* \rangle + \langle \mathbf{u}^*, \mathbf{d}^v \rangle + \langle \mathbf{d}^u, \mathbf{d}^v \rangle) \leq 2(\|\mathbf{d}^u\| + \|\mathbf{d}^v\|)$. Lemma now follows by combining the above observations with (25). $\qquad\square$

Now, we bound the third error term in (17). Note that although the two individual terms $((\langle \mathbf{u}_1, \mathbf{u}_2^* \rangle^2 B - F)\mathbf{u}_2^*$ and $(\langle \mathbf{u}_1, \mathbf{u}_2 \rangle^2 B - G)\mathbf{v})$ are both small, still it is critical to bound the difference as the individual terms can be as large as a constant, even when $\mathbf{u}_1 = \mathbf{u}_1^*$ and $\mathbf{u}_2 = \mathbf{u}_2^*$. However, the difference goes down linearly with $\|\mathbf{u}_2 - \mathbf{u}_2^*\|_2$.

**Lemma B.4.** *Let $B, C, F, G$ be defined as in* (18)*. Also, let the assumptions of Theorem B.1 hold. Also, let $p \geq C \frac{\mu^3 \log^2 n}{\gamma^2 n^{3/2}}$, where $C > 0$ is a global constant. Then, the following holds with probability $\geq 1 - 4/n^9$:*

$$\|(\langle \mathbf{u}_1, \mathbf{u}_2^* \rangle^2 B - F)\mathbf{u}_2^* - (\langle \mathbf{u}_1, \mathbf{u}_2 \rangle^2 B - G)\mathbf{u}_2\|_2 \leq 8\gamma \|\mathbf{u}_2 - \mathbf{u}_2^*\|_2.$$

*Proof.* Let $\mathbf{u}_2 = \mathbf{u}_2^* + \mathbf{d}_2^u$ and $\mathbf{u}_1 = \mathbf{u}_1^* + \mathbf{d}_1^u$. Then,

$$(\langle \mathbf{u}_1, \mathbf{u}_2 \rangle^2 B - G)\mathbf{u}_2 = (\langle \mathbf{u}_1, \mathbf{u}_2 \rangle^2 B - G)\mathbf{u}_2^* + (\langle \mathbf{u}_1, \mathbf{u}_2 \rangle^2 B - G)\mathbf{d}_2^u. \qquad (26)$$

Now,

$$G_{ii} = \sum_{jk} \delta_{ijk} \mathbf{u}_1(j)\mathbf{u}_1(k)\mathbf{u}_2(j)\mathbf{u}_2(k) = \sum_{jk} \delta_{ijk} \mathbf{u}_1(j)\mathbf{u}_1(k)(\mathbf{u}_2^*(j) + \mathbf{d}_2^u(j))(\mathbf{u}_2^*(k) + \mathbf{d}_2^u(k))$$

$$= \sum_{jk} \delta_{ijk} \mathbf{u}_1(j)\mathbf{u}_1(k) \left(\mathbf{u}_2^*(j)\mathbf{u}_2^*(k) + \mathbf{d}_2^u(j)\mathbf{u}_2^*(k) + \mathbf{u}_2^*(j)\mathbf{d}_2^u(k) + \mathbf{d}_2^u(j)\mathbf{d}_2^u(k)\right) = F_{ii} + D_{ii}^1 + D_{ii}^2 + D_{ii}^3.$$

$$(27)$$

Hence, using (24), and (27), we have:

$$(\langle \mathbf{u}_1, \mathbf{u}_2 \rangle^2 B - G) = (\langle \mathbf{u}_1, \mathbf{u}_2^* \rangle^2 B - F) + (\langle \mathbf{u}_1, \mathbf{u}_2^* \rangle \langle \mathbf{u}_1, \mathbf{d}_2^u \rangle B - D^1)$$
$$+ (\langle \mathbf{u}_1, \mathbf{u}_2^* \rangle \langle \mathbf{u}_1, \mathbf{d}_2^u \rangle B - D^2) + (\langle \mathbf{u}_1, \mathbf{d}_2^u \rangle^2 B - D^3) \quad (28)$$

Combining the above equation with (26), we get:

$$(\langle \mathbf{u}_1, \mathbf{u}_2 \rangle^2 B - G)\mathbf{u}_2 - (\langle \mathbf{u}_1, \mathbf{u}_2^* \rangle^2 B - F)\mathbf{u}_2^* = (\langle \mathbf{u}_1, \mathbf{u}_2^* \rangle \langle \mathbf{u}_1, \mathbf{d}_2^u \rangle B - D^1)\mathbf{u}_2^* + (\langle \mathbf{u}_1, \mathbf{u}_2^* \rangle \langle \mathbf{u}_1, \mathbf{d}_2^u \rangle B - D^2)\mathbf{u}_2^*$$
$$+ (\langle \mathbf{u}_1, \mathbf{d}_2^u \rangle^2 B - D^3)\mathbf{u}_2^* - (\langle \mathbf{u}_1, \mathbf{u}_2 \rangle^2 B - G)\mathbf{d}_2^u. \quad (29)$$

Lemma now follows using Lemma B.7, B.8, and the above equation. $\qquad\square$

We now present a few technical lemmas that are critical to our proofs of the above given lemmas.

**Lemma B.5.** *Let $\mathbf{u}, \mathbf{u}^* \in \mathbb{R}^n$ be $\mu$-incoherent unit vectors. Also, let $\delta_{jk}, 1 \leq j \leq n, 1 \leq k \leq n$ be i.i.d. Bernoulli random variables with $\delta_{jk} = 1$ w.p. $p \geq C\mu^4 \log^3 n/(\gamma^2 \cdot n^2)$.*

*Then, the following holds with probability $\geq 1 - 1/n^{10}$:*

$$|\frac{1}{p} \sum_{jk} \delta_{jk} \mathbf{u}(j)\mathbf{u}^*(j)\mathbf{u}(k)\mathbf{u}^*(k) - \langle \mathbf{u}, \mathbf{u}^* \rangle^2| \leq \gamma.$$

*where $\gamma \leq C/\log n$, where $C > 0$ is a global constant.*

**Lemma B.6.** *Let $\mathbf{u} \in \mathbb{R}^n$ be $\mu$-incoherent unit vectors. Also, let $a, b \in \mathbb{R}^n$ be s.t. $|a_i| \leq \frac{\mu}{\sqrt{n}}$ and $\|a\|_2 \leq 1$. Also, let $\delta_{jk}, 1 \leq j \leq n, 1 \leq k \leq n$ be i.i.d. Bernoulli random variables with $\delta_{jk} = 1$ w.p. $p \geq \frac{C\mu^3}{\gamma^2 n^{1.5}}$.*

*Then, the following holds with probability $\geq 1 - 1/n^{10}$:*

$$|\frac{1}{p} \sum_{jk} \delta_{jk} \mathbf{u}(j)a(j)\mathbf{u}(k)b(k) - \langle \mathbf{u}, a \rangle \langle \mathbf{u}, b \rangle| \leq \gamma \|b\|_2.$$

*where $\gamma \leq C/\log n$, where $C > 0$ is a global constant.*

**Lemma B.7.** *Let $\mathbf{u}$ be a fixed unit vector and let $a, b, c$ be fixed vectors in $\mathbb{R}^n$. Also, let all $\mathbf{u}, a, b, c \in \mathbb{R}^n$ be s.t. their $L_\infty$ norm is bounded by $\frac{\mu}{\sqrt{n}}$ and $L_2$ norm is bounded by 1. Also, let $p \geq \frac{C\mu^3(\log^2 n)}{\gamma^2 \cdot n^{3/2}}$, where $C > 0$ is a global constant. Then the following holds (w.p. $\geq 1 - 2/n^{10}$):*

$$\|(\langle \mathbf{u}, a\rangle\langle \mathbf{u}, b\rangle B - R)c\|_2 \leq \gamma\sqrt{1 - \langle \mathbf{u}, a\rangle^2\langle \mathbf{u}, b\rangle^2},$$

*where $B, R$ are both diagonal matrices with $B(i,i) = \frac{1}{p}\sum_{jk}\delta_{ijk}(\mathbf{u}(j))^2(\mathbf{u}(k))^2$, and $R(i,i) = \frac{1}{p}\sum_{jk}\delta_{ijk}\mathbf{u}(j)\mathbf{u}(k)a(j)b(k)$.*

**Lemma B.8.** *Let $\mathbf{u}$ be a fixed unit vector and let $a, b$ be fixed vectors in $\mathbb{R}^n$. Also, let all $\mathbf{u}$ be $\mu$-incoherent unit vectors, and $a$ be such that $\|a\|_\infty \leq \frac{\mu}{\sqrt{n}}$ and $\|a\|_2 \leq 1$. Also, let $p \geq \frac{C\mu^3(\log^2 n)}{\gamma^2 \cdot n^{3/2}}$, where $C > 0$ is a global constant. Then the following holds (w.p. $\geq 1 - 2/n^{10}$):*

$$\|(\langle \mathbf{u}, a\rangle\langle \mathbf{u}, b\rangle B - R)\|_2 \leq 2\gamma\|b\|_2$$

*where $B, R$ are diagonal matrices s.t. $B(i,i) = \frac{1}{p}\sum_{jk}\delta_{ijk}(\mathbf{u}(j))^2(\mathbf{u}(k))^2$, $R(i,i) = \frac{1}{p}\sum_{jk}\delta_{ijk}\mathbf{u}(j)\mathbf{u}(k)a(j)b(k)$.*

## B.3 Proofs of Technical Lemmas

*Proof of Lemma B.5.* Let $X_{jk} = \frac{1}{p}\delta_{jk}\mathbf{u}(j)\mathbf{u}^*(j)\mathbf{u}(k)\mathbf{u}^*(k)$. Note that, $|X_{jk}| \leq \frac{\mu^4}{pn^2}$. Also,

$$\mathbb{E}[\sum_{jk}X_{jk}^2] = \frac{1}{p}\sum_{jk}(\mathbf{u}(j))^2(\mathbf{u}^*(j))^2(\mathbf{u}(k))^2(\mathbf{u}^*(k))^2 \leq \frac{\mu^4}{pn^2}.$$

Hence, using Bernstein's inequality, we have:

$$Pr(|\sum_{jk}X_{jk} - \mathbb{E}[\sum_{jk}X_{jk}]| > t) \leq \exp(-\frac{pn^2}{\mu^4}\cdot\frac{t^2/2}{1 + t/3}).$$

Lemma now follows by selecting $t = C/\log n$. $\qquad\square$

*Proof of Lemma B.6.* Let $X_{jk} = \frac{1}{p}\delta_{jk}\mathbf{u}(j)a(j)\mathbf{u}(k)b(k)$. Note that, $|X_{jk}| \leq \frac{\mu^3\|b\|_2}{pn^{1.5}}$. Also,

$$\mathbb{E}[\sum_{jk}X_{jk}^2] = \frac{1}{p}\sum_{jk}(\mathbf{u}(j))^2(a(j))^2(\mathbf{u}(k))^2(b(k))^2 \leq \frac{\mu^4\|b\|^2}{pn^2} \leq \frac{\mu^3\|b\|^2}{pn^{1.5}}.$$

Hence, using Bernstein's inequality, we have:

$$Pr(|\sum_{jk}X_{jk} - \mathbb{E}[\sum_{jk}X_{jk}]| > t) \leq \exp(-\frac{pn^{1.5}}{\mu^3}\cdot\frac{t^2/2}{\|b\|_2^2 + \|b\|_2 t/3}).$$

Lemma now follows by selecting $t = \gamma\|b\|_2$. $\qquad\square$

*Proof of Lemma B.7.*

$$(\langle \mathbf{u}, a\rangle\langle \mathbf{u}, b\rangle B - R)c = \frac{1}{p}\sum_{ijk}\delta_{ijk}c_i(\langle \mathbf{u}, a\rangle\langle \mathbf{u}, b\rangle(u(j))^2(u(k))^2 - u(j)u(k)a(j)b(k))\mathbf{e}_i = \sum_{ijk}Z_{ijk},$$

(30)

where $Z_{ijk} = \frac{1}{p}\delta_{ijk}c_i(\langle \mathbf{u}, a\rangle\langle \mathbf{u}, b\rangle(u(j))^2(u(k))^2 - u(j)u(k)a(j)b(k))\mathbf{e}_i$. Note that,

$$\|Z_{ijk} - \mathbb{E}[Z_{ijk}]\|_2 \leq \frac{2}{p}c_i u(j)u(k)\sqrt{1 - \langle \mathbf{u}, a\rangle^2\langle \mathbf{u}, b\rangle^2} \leq \gamma\sqrt{1 - \langle \mathbf{u}, a\rangle^2\langle \mathbf{u}, b\rangle^2},$$

as $p \geq \frac{C\mu^3(\log^2 n)}{\gamma \cdot n^{3/2}}$. Also,

$$\|\sum_{ijk}\mathbb{E}[Z_{ijk}^T Z_{ijk}]\|_2 = \|\frac{1}{p}\sum_{ijk}c_i^2(u(j))^2(u(k))^2(\langle \mathbf{u}, a\rangle\langle \mathbf{u}, b\rangle u(j)u(k) - a(j)b(k))^2\|_2 \leq \frac{1}{p}\frac{\mu^4}{n^2}(1 - \langle \mathbf{u}, a\rangle^2\langle \mathbf{u}, b\rangle^2).$$

Hence, for $p$ and $\gamma$ mentioned above, we have:

$$\|\sum_{ijk} \mathbb{E}[Z_{ijk}^T Z_{ijk}]\|_2 \leq \gamma(1 - \langle \mathbf{u}, a \rangle^2 \langle \mathbf{u}, b \rangle^2).$$

Lemma now follows by using Bernstein's inequality and the fact that $\sum_{ijk} Z_{ijk} = 0$. $\qquad\square$

*Proof of Lemma B.8.* Consider the $i$-th element of the diagonal matrix $(\langle \mathbf{u}, a \rangle \langle \mathbf{u}, b \rangle B - R) = \langle \mathbf{u}, a \rangle \langle \mathbf{u}, b \rangle B(i, i) - R(i, i)$. Now, using Lemma B.5, $|B(i, i)| \leq 1 + \gamma$ w.p. $\geq 1 - 1/n^{10}$. Similarly, using Lemma B.6, $|R(i, i) - \langle \mathbf{u}, a \rangle \langle \mathbf{u}, b \rangle| \leq \gamma \|b\|_2$. Hence, w.p. $\geq 1 - 1/n^{10}$, we have:

$$|\langle \mathbf{u}, a \rangle \langle \mathbf{u}, b \rangle B(i, i) - R(i, i)| \leq 2\gamma \|b\|_2.$$

Lemma now follows by observing that
$\|(\langle \mathbf{u}, a \rangle \langle \mathbf{u}, b \rangle B - R)\|_2 = \max_i |\langle \mathbf{u}, a \rangle \langle \mathbf{u}, b \rangle B(i, i) - R(i, i)|$ and using the above mentioned bound with union bound. $\qquad\square$

## B.4 Proof of Theorem 2.3 and general rank-$r$ analysis of alternating minimization

*Proof.* We prove the theorem by showing the following for all $q$:

$$\sigma_q^* \left( (\Delta_q^\sigma)^{t+1} + \|\mathbf{d}_q^{t+1}\|_2 \right) \leq \frac{1}{2} d_\infty([U, \Sigma], [U^*, \Sigma^*]).$$

The update for $\widehat{\mathbf{u}}_q^{t+1}$ is given by:

$$\widehat{\mathbf{u}}_q^{t+1}(i) = \frac{\sum_{jk} \delta_{ijk} \sigma_q^* \cdot \mathbf{u}_q(j) \mathbf{u}_q(k) \mathbf{u}_q^*(j) \mathbf{u}_q^*(k)}{\sum_{jk} \delta_{ijk} \mathbf{u}_q(j)^2 \mathbf{u}_q(k)^2} \mathbf{u}_q^*(i)$$

$$+ \frac{\sum_{\ell \neq q} \sum_{jk} \delta_{ijk} \mathbf{u}_q(j) \mathbf{u}_q(k) (\sigma_\ell^* \cdot \mathbf{u}_\ell^*(i) \mathbf{u}_\ell^*(j) \mathbf{u}_\ell^*(k) - \sigma_\ell \cdot \mathbf{u}_\ell(i) \mathbf{u}_\ell(j) \mathbf{u}_\ell(k))}{\sum_{jk} \delta_{ijk} \mathbf{u}_q(j)^2 \mathbf{u}_q(k)^2}. \quad (31)$$

It can be written as a vector update,

$$\widehat{\mathbf{u}}_q^{t+1} = \sigma_q^* \langle \mathbf{u}_q, \mathbf{u}_q^* \rangle^2 \mathbf{u}_q^* - B^{-1}(\sigma_q^* \langle \mathbf{u}_q, \mathbf{u}_q^* \rangle^2 B - \sigma_q^* C) \mathbf{u}_q^* + \sum_{\ell \neq q} \left( \sigma_\ell^* \langle \mathbf{u}_q, \mathbf{u}_\ell^* \rangle^2 \mathbf{u}_\ell^* - \sigma_\ell \langle \mathbf{u}_q, \mathbf{u}_\ell \rangle^2 \mathbf{u}_\ell \right)$$

$$+ \sum_{\ell \neq q} B^{-1} \left( \sigma_\ell^* \cdot (\langle \mathbf{u}_q, \mathbf{u}_\ell^* \rangle^2 B - F_\ell) \mathbf{u}_\ell^* - \sigma_\ell \cdot (\langle \mathbf{u}_q, \mathbf{u}_\ell \rangle^2 B - G_\ell) \mathbf{u}_\ell \right),$$

$$(32)$$

where $B$, $C$, $F_\ell$, $G_\ell$ are all diagonal matrices, s.t.,

$$B(i, i) = \sum_{jk} \delta_{jk} \mathbf{u}_q(j)^2 \mathbf{u}_q(k)^2, \ C(i, i) = \sum_{jk} \delta_{jk} \mathbf{u}_q(j) \mathbf{u}_q^*(j) \mathbf{u}_q(k) \mathbf{u}_q^*(k),$$

$$F_\ell(i, i) = \sum_{jk} \delta_{ijk} \mathbf{u}_q(j) \mathbf{u}_q(k) \mathbf{u}_\ell^*(j) \mathbf{u}_\ell^*(k), \text{ and } G_\ell(i, i) = \sum_{jk} \delta_{ijk} \mathbf{u}_q(j) \mathbf{u}_q(k) \mathbf{u}_\ell(j) \mathbf{u}_\ell(k). \quad (33)$$

We decompose the error terms $\widehat{\mathbf{u}}_q^{t+1} - \sigma_q^* \mathbf{u}_q^* = \text{err}_q^0 + \sum_{\ell \neq q} (\text{err}_\ell^1 + \text{err}_\ell^2)$ and provide upper bounds for each, where

$$\text{err}_q^0 \equiv \sigma_q^* (\langle \mathbf{u}_q, \mathbf{u}_q^* \rangle^2 - 1) \mathbf{u}_q^* - \sigma_q^* B^{-1}(\langle \mathbf{u}_q, \mathbf{u}_q^* \rangle^2 B - C) \mathbf{u}_q^*,$$

$$\text{err}_\ell^1 \equiv \sigma_\ell^* \langle \mathbf{u}_q, \mathbf{u}_\ell^* \rangle^2 \mathbf{u}_\ell^* - \sigma_\ell \langle \mathbf{u}_q, \mathbf{u}_\ell \rangle^2 \mathbf{u}_\ell,$$

$$\text{err}_\ell^2 \equiv B^{-1} \left( \sigma_\ell^* \cdot (\langle \mathbf{u}_q, \mathbf{u}_\ell^* \rangle^2 B - F_\ell) \mathbf{u}_\ell^* - \sigma_\ell \cdot (\langle \mathbf{u}_q, \mathbf{u}_\ell \rangle^2 B - G_\ell) \mathbf{u}_\ell \right). \quad (34)$$

Using Lemma B.7, we have for all $p$ satisfying $p \geq (C\mu^3 (\log n)^2)/(\gamma^2 \, n^{3/2})$, with probability at least $1 - 2/n^{10}$:

$$\|\text{err}_q^0\|_2 \leq \sigma_q^* \left( \sqrt{1 - \langle \mathbf{u}_q, \mathbf{u}_q^* \rangle^2} + 2\gamma \right) \sqrt{1 - \langle \mathbf{u}_q, \mathbf{u}_q^* \rangle^2} \leq \sigma_q^* (\|\mathbf{d}_q\|_2 + 2\gamma) \|\mathbf{d}_q\|_2. \quad (35)$$

Eventually, we set $\gamma \leq \frac{1}{1600r} \cdot \frac{\sigma_{min}^*}{\sigma_{max}^*}$ to prove the theorem. Using Lemma B.10, we have (w.p. $\geq 1 - 1/n^8$):

$$\sum_{\ell \neq q} \|\text{err}_\ell^1\|_2 \leq 8 \sum_{\ell \neq q} (\|\mathbf{d}_q\|_2 + \|\mathbf{d}_\ell\|_2) \cdot \sigma_\ell^* \cdot (\|\mathbf{d}_\ell\|_2 + \Delta_\ell^\sigma). \tag{36}$$

Using Lemma B.11, we get (w.p. $\geq 1 - 1/n^8$):

$$\sum_{\ell \neq q} \|\text{err}_\ell^2\|_2 \leq 16\gamma \sum_{\ell \neq q} \sigma_\ell^* \cdot (\Delta_\ell^\sigma + \|\mathbf{d}_\ell\|_2). \tag{37}$$

Using (31), (35), (36), (37), we have (w.p. $\geq 1 - 3/n^8$):

$$\widehat{\mathbf{u}}_q^{t+1} = \sigma_q^{t+1} \mathbf{u}_q^{t+1} = \sigma_q^* \mathbf{u}_q^* + \text{err}_q, \tag{38}$$

where,

$$\|\text{err}_q\|_2 \leq \sigma_q^* (\|\mathbf{d}_\ell\|_2 + 2\gamma) \|\mathbf{d}_q\|_2 + 8 \sum_{\ell \neq q} (\|\mathbf{d}_q\|_2 + \|\mathbf{d}_\ell\|_2 + 2\gamma) \sigma_\ell^* (\|\mathbf{d}_\ell\|_2 + \Delta_\ell^\sigma). \tag{39}$$

Now, since $\|\mathbf{d}_\ell\|_2 \leq \frac{1}{1600r} \cdot \frac{\sigma_{min}^*}{\sigma_{max}^*}, \forall \ell$, and $\gamma \leq \frac{1}{1600r}$, we have (w.p. $\geq 1 - 3/n^8$):

$$\|\text{err}_q\|_2 \leq \frac{\sigma_q^*}{16} \cdot \frac{\sigma_{min}^*}{\sigma_{max}^*} \|\mathbf{d}_q\|_2 + \frac{1}{16} \cdot \sigma_{min}^* \cdot d_\infty([U, \Sigma], [U^*, \Sigma^*]), \tag{40}$$

Using (38) and (40), and the fact that $|\sigma_q^{t+1} - \sigma_q^*| \leq |\sigma_q^{t+1} \mathbf{u}_q^{t+1} - \mathbf{u}_q^* \sigma_q^*|$ for normalized vectors $\mathbf{u}_q^{t+1}$ and $\mathbf{u}_q^*$, we have:

$$|\sigma_q^{t+1} - \sigma_q^*| \leq \frac{\sigma_q^*}{16} \|\mathbf{d}_q\|_2 + \frac{\sigma_{min}^*}{16} d_\infty([U, \Sigma], [U^*, \Sigma^*]) \leq \frac{\sigma_q^*}{8} d_\infty([U, \Sigma], [U^*, \Sigma^*]). \tag{41}$$

Similarly, using (38) and (41), we have:

$$\sigma_q^* \|\mathbf{u}_q^{t+1} - \mathbf{u}_q^*\|_2 \leq \frac{\sigma_q^*}{4} d_\infty([U, \Sigma], [U^*, \Sigma^*]). \tag{42}$$

That is,

$$(\Delta_q^\sigma)^{t+1} + \|\mathbf{d}_q^{t+1}\|_2 \leq \frac{1}{2} d_\infty([U, \Sigma], [U^*, \Sigma^*]). \tag{43}$$

First part of the Theorem now follows by observing that $d_\infty([U^{t+1}, \Sigma^{t+1}], [U^*, \Sigma^*]) = \max_q \sigma_q^* ((\Delta_q^\sigma)^{t+1} + \|\mathbf{d}_q^{t+1}\|_2)$ and by using the above equation.

Second part of the Theorem follows directly from Lemma B.9.

$\square$

## B.5 Technical lemmas for general rank-$r$ analysis

**Lemma B.9.** *Let $\hat{\mathbf{u}}_q^{t+1}$ be obtained by update (31) and let $\mathbf{u}_q^{t+1} = \hat{\mathbf{u}}_q^{t+1}/\|\hat{\mathbf{u}}_q^{t+1}\|_2$. Also, let the conditions given in Theorem 2.3 hold. Then, w.p. $\geq 1 - 1/n^9$, $\mathbf{u}_q^{t+1}$ is $2\mu$-incoherent.*

*Proof.* Using (31) and the definitions of $B, C, F_\ell, G_\ell$ given in (33), we have:

$$|\hat{\mathbf{u}}_q^{t+1}(i)| \leq \sigma_q^* \frac{|C(i,i)|}{|B(i,i)|} \frac{\mu}{\sqrt{n}} + \sum_\ell \sigma_\ell^* \frac{|F_\ell(i,i)|}{|B(i,i)|} \frac{\mu}{\sqrt{n}} + \sigma_\ell \frac{|G_\ell(i,i)|}{|B(i,i)|} \frac{\mu}{\sqrt{n}},$$

$$\leq \left( \sigma_q^*(1+\gamma)/(1-\gamma) + \sum_\ell \sigma_\ell^*(\gamma + \|\mathbf{d}_\ell\|_2) + 2 \sum_\ell \sigma_\ell^* \cdot (1 + \Delta_\ell^\sigma) \cdot (\gamma + \|\mathbf{d}_\ell\|_2) \right) \mu/\sqrt{n},$$

$$\leq \sigma_q^*(1 + \frac{1}{100}) \cdot \frac{\mu}{\sqrt{n}} \tag{44}$$

where the second inequality follows by bounds on $B_{ii}, C_{ii}, F_{ii}, G_{ii}$ obtained using Lemma B.6 and the distance bound $d_\infty([\mathbf{u}_1, \mathbf{u}_2], [\mathbf{u}_1^*, \mathbf{u}_2^*])$.

Lemma now follows using (44) and the bound on $|\sigma_q^{t+1} - \sigma_q^*|$ given by (43). $\square$

**Lemma B.10.** *Let* $\mathbf{d}_\ell = \mathbf{u}_\ell^* - \mathbf{u}_\ell$ *and* $\Delta_\ell^\sigma = |\sigma_\ell - \sigma_\ell^*|/\sigma_\ell^*$, *where* $\|\mathbf{d}_\ell\|_2 \leq 1$. *Let* $\mathbf{u}_\ell^*, \mathbf{u}_\ell, \forall 1 \leq \ell \leq r$ *be unit vectors and let* $\langle \mathbf{u}_\ell^*, \mathbf{u}_q^* \rangle = 0, \forall \ell \neq q$. *Then, the following holds:*

$$\|\sigma_\ell^* \cdot \langle \mathbf{u}_q, \mathbf{u}_\ell^* \rangle^2 \mathbf{u}_\ell^* - \sigma_\ell \cdot \langle \mathbf{u}_q, \mathbf{u}_\ell \rangle^2 \mathbf{u}_\ell\| \leq 4\sigma_\ell^*(\|\mathbf{d}_\ell\|_2 + \|\mathbf{d}_q\|_2)(\|\mathbf{d}_\ell\|_2 + \Delta_\ell^\sigma).$$

*Proof.* Let $\sigma_\ell = \sigma_\ell^* + \Delta_\ell^\sigma$.

Now,

$$\langle \mathbf{u}_q, \mathbf{u}_\ell \rangle^2 = (\langle \mathbf{u}_q, \mathbf{u}_\ell^* \rangle + \langle \mathbf{u}_q, \mathbf{d}_\ell \rangle)^2 = \langle \mathbf{u}_q, \mathbf{u}_\ell^* \rangle^2 + \langle \mathbf{u}_q, \mathbf{d}_\ell \rangle^2 + 2\langle \mathbf{u}_q, \mathbf{u}_\ell^* \rangle \langle \mathbf{u}_q, \mathbf{d}_\ell \rangle. \tag{45}$$

Hence,

$$\|\langle \mathbf{u}_q, \mathbf{u}_\ell^* \rangle^2 \mathbf{u}_\ell^* - \langle \mathbf{u}_q, \mathbf{u}_\ell \rangle^2 \mathbf{u}_\ell\|_2 = \|\langle \mathbf{u}_q, \mathbf{u}_\ell \rangle^2 \mathbf{d}_\ell - \langle \mathbf{u}_q, \mathbf{d}_\ell \rangle^2 \mathbf{u}_\ell^* - 2\langle \mathbf{u}_q, \mathbf{u}_\ell^* \rangle \langle \mathbf{u}_q, \mathbf{d}_\ell \rangle \mathbf{u}_\ell^*\|_2,$$
$$\leq \langle \mathbf{u}_q, \mathbf{u}_\ell \rangle^2 \|\mathbf{d}_\ell\|_2 + \|\mathbf{d}_\ell\|^2 + 2|\langle \mathbf{u}_q, \mathbf{u}_\ell^* \rangle|\|\mathbf{d}_\ell\|_2. \tag{46}$$

Now, $\langle \mathbf{u}_q, \mathbf{u}_\ell^* \rangle = \langle \mathbf{d}_q, \mathbf{u}_\ell^* \rangle \leq \|\mathbf{d}_q\|_2$. Also, $\langle \mathbf{u}_q, \mathbf{u}_\ell \rangle^2 \leq \langle \mathbf{u}_q, \mathbf{u}_\ell \rangle = (\langle \mathbf{d}_q, \mathbf{u}_\ell^* \rangle + \langle \mathbf{u}_q^*, \mathbf{d}_\ell \rangle + \langle \mathbf{d}_q, \mathbf{d}_\ell \rangle) \leq 2(\|\mathbf{d}_q\| + \|\mathbf{d}_\ell\|)$. Using the above observations with (46), we have:

$$\|\sigma_\ell^* \cdot \langle \mathbf{u}_q, \mathbf{u}_\ell^* \rangle^2 \mathbf{u}_\ell^* - \sigma_\ell \cdot \langle \mathbf{u}_q, \mathbf{u}_\ell \rangle^2 \mathbf{u}_\ell\| \leq \sigma_\ell^* \|\langle \mathbf{u}_q, \mathbf{u}_\ell^* \rangle^2 \mathbf{u}_\ell^* - \langle \mathbf{u}_q, \mathbf{u}_\ell \rangle^2 \mathbf{u}_\ell\|_2 + \sigma_\ell^* \cdot \Delta_\ell^\sigma \langle \mathbf{u}_q, \mathbf{u}_\ell \rangle^2.$$

Lemma now follows by combining the above equation with the above given bound on $\langle \mathbf{u}_q, \mathbf{u}_\ell \rangle^2$. $\square$

**Lemma B.11.** *Let* $\mathbf{u}_\ell$, $\mathbf{d}_\ell$, $\Delta_\ell^\sigma$, $\forall \ell$ *be as defined in Theorem 2.3 and let* $B, F_\ell, G_\ell$ *be as defined in* (33). *Also, let* $T$ *and* $p$ *satisfy assumptions of Theorem 2.3. Then, the following holds with probability* $\geq 1 - 4/n^9$:

$$\|\sigma_\ell^* \cdot (\langle \mathbf{u}_q, \mathbf{u}_\ell^* \rangle^2 B - F_\ell)\mathbf{u}_\ell^* - \sigma_\ell \cdot (\langle \mathbf{u}_q, \mathbf{u}_\ell \rangle^2 B - G_\ell)\mathbf{u}_\ell\|_2 \leq 8\gamma \cdot \sigma_\ell^* \cdot (\Delta_\ell^\sigma + \|\mathbf{d}_\ell\|_2).$$

*Proof.*
$$(\langle \mathbf{u}_q, \mathbf{u}_\ell \rangle^2 B - G_\ell)\mathbf{u}_\ell = (\langle \mathbf{u}_q, \mathbf{u}_\ell \rangle^2 B - G_\ell)\mathbf{u}_\ell^* + (\langle \mathbf{u}_q, \mathbf{u}_\ell \rangle^2 B - G_\ell)\mathbf{d}_\ell. \tag{47}$$

Now,

$$G_\ell(i,i) = \sum_{jk} \delta_{ijk} \mathbf{u}_q(j)\mathbf{u}_q(k)\mathbf{u}_\ell(j)\mathbf{u}_\ell(k) = \sum_{jk} \delta_{ijk} \mathbf{u}_q(j)\mathbf{u}_q(k)(\mathbf{u}_\ell^*(j) + \mathbf{d}_\ell(j))(\mathbf{u}_\ell^*(k) + \mathbf{d}_\ell(k))$$

$$= \sum_{jk} \delta_{ijk} \mathbf{u}_q(j)\mathbf{u}_q(k)(\mathbf{u}_\ell^*(j)\mathbf{u}_\ell^*(k) + \mathbf{d}_\ell(j)\mathbf{u}_\ell^*(k) + \mathbf{u}_\ell^*(j)\mathbf{d}_\ell(k) + \mathbf{d}_\ell(j)\mathbf{d}_\ell(k))$$

$$= F_\ell(i,i) + D^1(i,i) + D^2(i,i) + D^3(i,i). \tag{48}$$

Using (45), and (48), we have:

$$(\langle \mathbf{u}_q, \mathbf{u}_\ell \rangle^2 B - G_\ell) = (\langle \mathbf{u}_q, \mathbf{u}_\ell^* \rangle^2 B - F_\ell) + (\langle \mathbf{u}_q, \mathbf{u}_\ell^* \rangle \langle \mathbf{u}_q, \mathbf{d}_\ell \rangle B - D^1)$$
$$+ (\langle \mathbf{u}_q, \mathbf{u}_\ell^* \rangle \langle \mathbf{u}_q, \mathbf{d}_\ell \rangle B - D^2) + (\langle \mathbf{u}_q, \mathbf{d}_\ell \rangle^2 B - D^3) \tag{49}$$

Combining the above equation with (47), we get:

$$(\langle \mathbf{u}_q, \mathbf{u}_\ell \rangle^2 B - G_\ell)\mathbf{u}_\ell - (\langle \mathbf{u}_q, \mathbf{u}_\ell^* \rangle^2 B - F_\ell)\mathbf{u}_\ell^* = (\langle \mathbf{u}_q, \mathbf{u}_\ell^* \rangle \langle \mathbf{u}_q, \mathbf{d}_\ell \rangle B - D^1)\mathbf{u}_\ell^* + (\langle \mathbf{u}_q, \mathbf{u}_\ell^* \rangle \langle \mathbf{u}_q, \mathbf{d}_\ell \rangle B - D^2)\mathbf{u}_\ell^*$$
$$+ (\langle \mathbf{u}_q, \mathbf{d}_\ell \rangle^2 B - D^3)\mathbf{u}_\ell^* - (\langle \mathbf{u}_q, \mathbf{u}_\ell \rangle^2 B - G_\ell)\mathbf{d}_\ell. \tag{50}$$

Hence, using Lemma B.7 and B.8, we get:

$$\sigma_\ell^* \|(\langle \mathbf{u}_q, \mathbf{u}_\ell \rangle^2 B - G_\ell)\mathbf{u}_\ell - (\langle \mathbf{u}_q, \mathbf{u}_\ell^* \rangle^2 B - F_\ell)\mathbf{u}_\ell^*\|_2 \leq 8\gamma\sigma_\ell^* \|\mathbf{d}_\ell\|_2. \tag{51}$$

Similarly, using Lemma B.7, we have:

$$\Delta_\ell^\sigma \cdot \sigma_\ell^* \|(\langle \mathbf{u}_q, \mathbf{u}_\ell \rangle^2 B - G_\ell)\mathbf{u}_\ell\|_2 \leq \gamma \cdot \Delta_\ell^\sigma \cdot \sigma_\ell^*. \tag{52}$$

Lemma now follows by combining (51), (52), and by using triangular inequality. $\square$

## B.6 Proof of Lemma 2.4

*Proof.*

$$\|\sigma_a(u_a \otimes u_a \otimes u_a) - \sigma_a^*(u_a^* \otimes u_a^* \otimes u_a^*)\|_F$$

$$\leq \quad \tilde{\varepsilon}\sigma_a^* + \sigma_a^*\|(u_a \otimes u_a \otimes u_a) - (u_a^* \otimes u_a^* \otimes u_a^*)\|_F$$

$$\leq \quad \tilde{\varepsilon}\sigma_a^* + \sigma_a^*\Big(\|(u_a - u_a^*) \otimes u_a^* \otimes u_a^*\|_F + \|u_a \otimes (u_a - u_a^*) \otimes u_a^*\|_F - \|u_a \otimes u_a \otimes (u_a - u_a^*)\|_F\Big)$$

$$\leq \quad 4\,\tilde{\varepsilon}\,\sigma_a^* \,.$$

Similarly, applying Cauchy-Schwartz,we get for $a \neq b$,

$$\langle \sigma_a(u_a \otimes u_a \otimes u_a) - \sigma_a^*(u_a^* \otimes u_a^* \otimes u_a^*), \sigma_b(u_b \otimes u_b \otimes u_b) - \sigma_b^*(u_b^* \otimes u_b^* \otimes u_b^*)\rangle$$

$$\leq \quad 16\,\tilde{\varepsilon}^2\,\sigma_a^*\sigma_b^* \,.$$

It follows that

$$\|T - \widehat{T}\|_F^2 \quad = \quad \sum_{a,b\in[r]} \langle \sigma_a(u_a \otimes u_a \otimes u_a) - \sigma_a^*(u_a^* \otimes u_a^* \otimes u_a^*), \sigma_b(u_b \otimes u_b \otimes u_b) - \sigma_b^*(u_b^* \otimes u_b^* \otimes u_b^*)\rangle$$

$$\leq \quad 16\,\tilde{\varepsilon}^2\,(\sum_a \sigma_a^*)^2 \quad \leq \quad (4\,\tilde{\varepsilon}\,r\,\sigma_{\max}^*)^2 \quad \leq \quad (4\,\tilde{\varepsilon}\,r^{1/2}\,\|T\|_F)^2 \,.$$

$\square$