[Reviews · NeurIPS 2014]

Submitted by Assigned_Reviewer_8

The authors describe a novel alternating minimization type of algorithm for computing an orthogonal factorization of a given tensor where only a subset of its entries have been revealed (from a special probability distribution). The result in this paper extends similar results appeared before for matrices. This is a clear accept.
Summary: The authors describe a novel alternating minimization type of algorithm for computing an orthogonal factorization of a given tensor where only a subset of its entries have been revealed (from a special probability distribution). The result in this paper extends similar results appeared before for matrices. This is a clear accept.

Submitted by Assigned_Reviewer_11

The authors address the question of recovering a low CP-rank incoherent tensor from missing data. They prove that if enough elements are sampled, the tensor can be recovered exactly using an algorithm they describe. Such results were known for matrices, but tensors are conceptually more complex. The paper gives an extensive theoretical analysis. No experimental results.

While the bounds are rather complex, and probably not very informative in practice (e.g. you do not know the coherence of T or its norm...) the algorithm is fairly simple, which is definitely a big plus. However, in that respect I was a bit disappointed that the author chose not to implement the algorithm and experiment with it. Sure, their theoretical analysis seems rather extensive and by itself is a sufficient contribution. However, since this does not seem just an "on-paper" algorithm, the impact and importance of the paper would have been significantly enhanced if the authors would have chosen to go the extra mile.

I did not read the proofs in detail and check them. They are too extensive and complex to do so in the time allocated for this review. One thing that did bother me a bit, and I would like the authors to address in the rebuttal, is the orthogonality of the u vectors found by the algorithm and by intermediate iterations. In the CP definition, the vectors u are orthogonal. But in algorithm and intermediate stages they are not. Is that an issue, and did you address that in the analysis? (for example, in the proof of Lemma B.2, isn't there an assumption that u_1^t+1 is orthogonal to other u's?)
Summary: Novel algorithm and analysis for tensor completion of low CP-rank incoherent tensors. Impressive theoretical analysis, but disappointingly no experimental evaluation.

Submitted by Assigned_Reviewer_31

This paper proposes an algorithm for the exact recovery of low-rank tensors from randomly chosen O(r^5n^{3/2}log^4(n)) entries (in the case of rank-r symmetric n x n x n tensor).

The proposed algorithm uses the robust tensor power method of Anandkumar et al. [1] to obtain an inital solution for the least squares iteration, which is a popular approach for tensor factorization. The proof relies on a rather strong notion of incoherence of the factor matrix and the proposed algorithm relies on the knowledge of the correct rank and the incoherence parameter.

The theoretically predicted scaling is confirmed in a synthetic experiment except that the authors heuristically assume a milder dependency with respect to the rank r (r^{1/2} instead of r^5).

The strength of the paper is that as far as I know, this is the first paper that presents a tensor completion algorithm with a O(n^{3/2}) guarantee from randomly chosen entries, though the authors have missed two related studies: Krishnamurthy & Singh (NIPS 2013) have shown a O(n) sample complexity when the samples are adaptively chosen; Tomioka et al. (NIPS 2011) have analyzed the random design setting and have shown a O(n^2) sample complexity.

Another interesting point is that it uses the robust tensor power method as an initalizer for a more conventional iterative algorithm, which have shown to be an effective strategy by other studies (Chaganty & Liang, ICML 2013).

On the negative side, the incoherence assumption employed by the authors seems rather strong. It basically ensures that all the entries of the true tensor T have the magnitude of O(n^{-3/2}). Compared to the fact that it is possible to show a tight sample complexity when the entries have magnitude O(1) in matrix completion (Foygel & Srebro, COLT2011), this is rather disappointing.

The algorithm also assumes the knowledge of the correct rank r and the incoherence parameter mu. It was not clear to me what happens if these parameters were mis-specified.

Minor comments:
- sigma_max and sigima_min whould be defined when they first appear (in page 3).
- I am not sure if the claim "the above theorem implies that among n^2 fibers of the form ... it is sufficient to have only O(n^{3/2}) fibers with any samples" (page 4) is true. At least it is not implied by the theorem because the theorem assumes uniform sampling. For example if only the first entries for all 3n^2 fibers were sampled, would this be enough to recover any incoherent low-rank tensor? This is not true for r > 1 for matrix completion because removing any observed entry would make the graph disconnected (see Kiraly and Tomioka, ICML 2012).
- The claim that the degrees of freedom of a symmetric 3rd order tensor is rn-r^2 (page 4) needs a proof or a reference.
Summary: Low-rank tensor completion algorithm that requires O(n^{3/2}) samples. The assumptions are rather strong.
Author Feedback
Author rebuttal: Addressing Reviewer_11's comments:

We want to emphasize that the proposed algorithm is designed with complexity and implementation issues in mind. We believe that the impact and the importance of the paper is greatly enhanced by the fact that we have implemented the algorithm, experimented with it, and made it publicly available. In the submitted manuscript, we have run numerical experiments with this implementation and the results are shown in Figures 1 and 2. The figures show that our algorithm is fast (with linear convergence as proved by our theorem) and our analysis is quite close to how the algorithm actually performs (in terms of the dependence in $n$).

We DO NOT assume that in the intermediate step the estimated u^t's are orthogonal to other `singular vectors'. We only assume that in the first step, we have a good estimate such that it is sufficiently close to the true singular vector (which can be done easily using Robust Tensor Power Method of [1]). We want to emphasize that in the analysis (including the proof of Lemma B.2) we do not require the intermediate estimates to be orthogonal to the other `singular vectors'. One of the main contribution in the analysis is that we show that the algorithm can correct for the error due to this `non-orthogonality’ of the intermediate steps (along with errors due to the missing entries, and the errors in the initial guess) and prove convergence to the true vectors.

Addressing Reviewer_31's comments:

We would like to thank the reviewer for pointing out the missed references, which will be added in the final version. All the minor comments from the reviewer will be addressed in the final version of the paper, and the claim in page 4 that "the above theorem implies..." will be made more clear.

In the numerical experiments, we omitted the clipping step and did not make the true incoherence $mu$ available to the algorithm (which will be explained clearly in the experimental results section). The simulation results suggests that the algorithm is not sensitive to the knowledge of $\mu$.

The incoherence assumption itself does not impose any constraints on the magnitude of the tensor entry, since the `singular values’ of the tensor will rescale the tensor entries. Therefore, when we compare the guarantees of the normalized error (and with of course the same ground truth tensors), then Theorem 2.1 of our manuscript and the main result from Foygel & Srebro 2011, achieve comparable error guarantees under the same assumptions (with no dependence on the incoherence). Further our analysis is slightly better by a poly-logarithmic factor.

If we use the analysis of Theorem 2.1 to give a similar guarantee for a (symmetric) matrix, we get that normalized error is smaller than \varepsilon, i.e. $(1/n^2)\| T- \hT\|_F^2 \leq \varepsilon$, if the sample size is $O((rn/\varepsilon) )$. The main result of Foygel&Srebro guarantees the same error for sample size O((rn/\varepsilon)\log(1/\varepsilon)^3). Hence, Theorem 2.1 provides a comparable bound on the sample complexity, regardless of what the incoherence of the original matrix is. Further, the main theorem of Theorem 1.1 should be thought of as providing even tighter guarantee. The incoherence of the tensor is not a strong assumptions, instead should be considered as a metric on the original tensor measuring how difficult it is to complete the tensor from sampled entries.

Addressing Reviewer_8's comments:

We thank the reviewer for detailed reading and comments.